# CMI-RewardBench: Evaluating Music Reward Models with Compositional Multimodal Instruction

**Yinghao Ma** [1] [*]  **Haiwen Xia** [2] [*]  **Hewei Gao** [3] [4]  **Weixiong Chen** [1]  **Yuxin Ye** [5]  **Yuchen Yang** [6]  **Sungkyun Chang** [1]
**Mingshuo Ding** [2]  **Yizhi Li** [7]  **Ruibin Yuan** [8]  **Simon Dixon** [1]  **Emmanouil Benetos** [1]

## Abstract

While music generation models have evolved to handle complex multimodal inputs mixing text, lyrics, and reference audio, evaluation mechanisms have lagged behind. In this paper, we bridge this critical gap by establishing a comprehensive ecosystem for music reward modeling under **Compositional Multimodal Instruction (CMI)**, where the generated music may be conditioned on text descriptions, lyrics, and audio prompts. We first introduce **CMI-Pref-Pseudo**, a large-scale preference dataset comprising 110k pseudo-labeled samples, and **CMI-Pref**, a high-quality, human-annotated corpus tailored for fine-grained alignment tasks. To unify the evaluation landscape, we propose **CMI-RewardBench**, a unified benchmark that evaluates music reward models on heterogeneous samples across musicality, text–music alignment, and compositional instruction alignment. Leveraging these resources, we develop **CMI reward models (CMI-RMs)**, a parameter-efficient reward model family capable of processing heterogeneous inputs. We evaluate their correlation with human judgment scores on musicality and alignment on CMI-Pref along with previous datasets. Further experiments demonstrate that CMI-RM not only correlates strongly with human judgments, but also enables effective **inference-time scaling** via top-$k$ filtering. Code is available at GitHub. Model weights: CMI-RM. Datasets: CMI-Pref-Pseudo and CMI-Pref.

[*]Equal contribution  [1]Queen Mary University of London, London, UK [2]Peking University, Beijing, China [3]Technical University of Munich, Munich, Germany [4]Technical University of Denmark, Lyngby, Denmark [5]Beijing University of Post and Telecommunications, Beijing, China [6]Soochow University, Suzhou, China [7]University of Manchester, Manchester, UK [8]Hong Kong University of Science and Technology, Hong Kong, China. Correspondence to: Yinghao Ma <yinghao.ma@qmul.ac.uk>, Emmanouil Benetos <emmanouil.benetos@qmul.ac.uk>.

*Proceedings of the $43^{rd}$ International Conference on Machine Learning*, Seoul, South Korea. PMLR 306, 2026. Copyright 2026 by the author(s).

## 1. Introduction

The rapid advancement of Artificial Intelligence Generated Content (AIGC) has significantly impacted the creative industries, with music generation emerging as one of the cornerstones for numerous commercial applications in music, movie and entertainment industries (Cao et al., 2025; Ren et al., 2025; Ma et al., 2024). Despite the proliferation of sophisticated generative models, evaluating their outputs remains a fundamental challenge. Generally speaking, music evaluation requires assessing both musicality and instruction following. However, advancements in generative models now necessitate this assessment under flexible multimodal conditions, such as text-only, lyric-guided, and audio-referenced inputs.

Developing these evaluation models is hindered by a critical data scarcity. Although large-scale user interaction data exists in music recommendation (e.g., Spotify Million Playlist (Papreja et al., 2019)), it fundamentally captures user-item affinity—a global preference for genre styles or playlists—rather than generative alignment, which demands assessment of perceptual quality and precise instruction-following. Such recommendation datasets lack the fine-grained, comparative rankings of generated samples against complex, multimodal instructions (such as interwoven lyrics, text descriptions, and reference audio) required to train alignment models (Deshmukh et al., 2024).

Consequently, evaluation methodologies have struggled to bridge this gap. Traditional metrics like Fréchet Audio Distance (FAD) (Kilgour et al., 2019) operate at the distribution level, failing to provide the sample-level signals necessary for post-training or filtering. More recent approaches, such as SongEval (Yao et al., 2025), PAM (Deshmukh et al., 2024), and various MOS predictors (Tjandra et al., 2025), have advanced the field by offering sample-level scoring. However, these efforts remain fragmented and narrowly specialized. They typically focus on isolated attributes (e.g., only caption alignment) and rely on rigid input assumptions, whereas state-of-the-art music generation models already support flexible input combinations, ranging from simple text prompts to interwoven lyrics and audio references. This growing mismatch between model capabilities and evalua-

tion methodologies is highlighted in Figure 1.

We argue that effective evaluation requires compositional alignment, defined here not merely as adherence to simultaneous constraints, but as the capability of a unified model to adaptively agree with human preferences across these optional and varying input conditions. Specifically, the model should assign scores or rankings that consistently reflect human judgments regarding both musical quality and instruction adherence, regardless of whether inputs are text-only, lyric-guided, or audio-referenced. A framework capable of judging this versatility is currently missing.

To bridge this gap, our proposed **CMI-RewardBench** integrates diverse task-specific datasets to vigorously evaluate whether a single reward model can judge generation quality against the heterogeneous instruction sets inherent to modern AIGC flows.

In this paper, our contributions are three-fold:

1. We construct **CMI-Pref-Pseudo**, containing 110k samples labeled via a robust pipeline using Qwen3-Omni (Xu et al., 2025b) with consistency filtering. Complementing this, we introduce **CMI-Pref**, a high-quality corpus of 4,027 pairs annotated by 31 human experts. These annotations capture fine-grained preferences for musicality, alignment, and confidence levels across diverse genres, instruments, and multimodal prompts (including lyrics and audio-to-audio conditioning).

2. We propose **CMI-RewardBench**, a unified benchmark for music reward models. By integrating existing resources (PAM, MusicEval, Music Arena) with our CMI-Pref test split, this benchmark evaluates models on five distinct tasks ranging from absolute musicality scoring to complex compositional alignment. This unified approach serves as a rigorous testbed for model **versatility** across optional input settings. Our baseline evaluations on this benchmark expose a significant capability gap, revealing that even state-of-the-art multimodal LLMs (e.g., Gemini-2.5-Pro) struggle to exceed 80% agreement with human preferences.

3. We develop **CMI-RM**, a family of music reward models supporting compositional conditioning over text, lyrics, and audio. Uniquely supporting all evaluation settings in CMI-RewardBench via a single, parameter-efficient architecture (∼30M), CMI-RM achieves performance comparable to or better than specialized open-source baselines like SongEval. Furthermore, we demonstrate that CMI-RM provides measurable benefits when used for top-k filtering, enabling "inference-time scaling" for music generation.

## 2. Related Work

### 2.1. RLHF for LLMs and MLLMs

Reinforcement Learning from Human Feedback (RLHF) has successfully aligned generative models with human intent by replacing heuristic proxies (e.g., ROUGE) with learned reward models (Stiennon et al., 2020; Ouyang et al., 2022; Bai et al., 2022). This paradigm has successfully extended to multimodal domains, including text-to-image generation (Kirstain et al., 2023; Xu et al., 2023) and video synthesis (Ahn et al., 2024; Liu et al., 2026), to address aesthetic quality and semantic alignment. Recent research has also applied preference optimization to speech synthesis to improve naturalness and bridge inference-time distribution gaps (Zhang et al., 2024a; 2025b). Concurrently, the use of LLMs as scalable evaluators (Zheng et al., 2023; Gu et al., 2024) has emerged as a robust alternative to human experts. While these works address text, vision, and speech, music evaluation remains under-explored and fragmented. Current music metrics lack the capacity to judge condition on compositional instructions.

### 2.2. Evaluation Metrics of Music

Music evaluation typically bifurcated into distribution-level quality assessment and sample-level alignment metrics. However, existing approaches struggle to address the complexity of compositional instructions.

**Quality and Subjective Metrics.** Distributional metrics like the Fréchet Audio Distance (FAD) (Kilgour et al., 2019) are the standard for global corpus assessment, with recent variants like MAD (Huang et al., 2025b) and KAD (Chung et al., 2025) improving correlation with human perception. For sample-level musicality, MOS predictors such as PAM (Deshmukh et al., 2024), Audiobox (Tjandra et al., 2025) and SongEval (Yao et al., 2025) evaluate aesthetic quality. While effective for general audio, high-performing systems like MusicRL (Cideron et al., 2024), WhisQ (Emon et al., 2025), QAMRO (Wang et al., 2025a), and DRAGON (Bai et al., 2025) remain closed-source.

**Alignment and LLM Judging.** Alignment is primarily measured via contrastive scores like CLAP (Wu et al., 2023), CLaMP3 (Wu et al., 2025), and MuQ-Mulan (Zhu et al., 2025). While music-specific checkpoints improve human-preference alignment (Grötschla et al., 2025), these metrics are largely restricted to text-to-audio pairs and neglect lyrics or audio prompts. Besides, emerging "LLM-as-a-judge" frameworks like AutoMV (Tang et al., 2025) and music recommendation AutoRaters (Chen et al., 2024b) offer scalable, multimodal evaluation with complex instructions. However, these rely on proprietary models and lack an open-source framework for evaluating compositional multimodal music instructions with lyrics and audio prompts information.

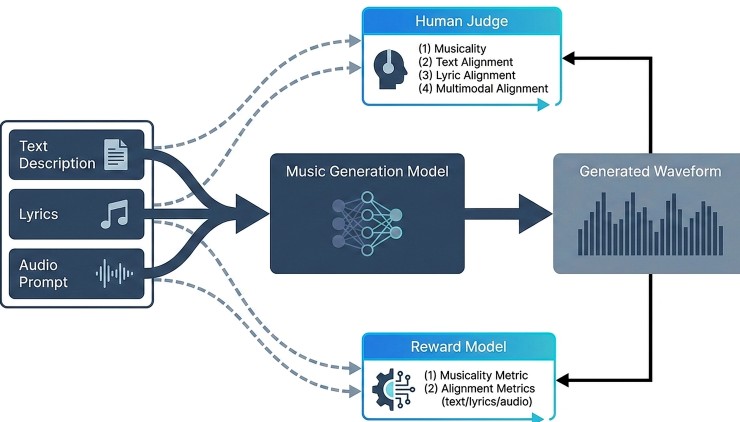

*Figure 1.* Reward models should act as proxies for human evaluation under compositional multimodal instructions (CMI). Human judges evaluate a generated waveform with respect to the provided prompt—text descriptions, lyrics, and/or reference audio—considering both musicality and instruction alignment. However, current reward models are typically fragmented: they either score musicality from audio alone or capture alignment for a single modality pair, and thus cannot reliably proxy human judgments across flexible CMI inputs.

## 2.3. Preference Datasets and Platforms

The development of robust reward models relies on standardized preference datasets and evaluation platforms. Early efforts such as MusicEval (Liu et al., 2025a), **SongEval** (Yao et al., 2025), and AudioEval (Wang et al., 2025b) provide expert-annotated corpora for absolute quality prediction and community benchmarks (Huang et al., 2025a; Ma et al., 2026; Zhang et al., 2025a). Recent pairwise datasets such as AIME (Grötschla et al., 2025) and MusicPref (Huang et al., 2025b) benchmark text-to-music systems via human comparisons, but remain limited to text-to-music generation.

Inspired by the crowdsourcing success of Chatbot Arena (Chiang et al., 2024), GenAI Arena (Jiang et al., 2024), and Copilot Arena (Chi et al., 2025) etc., the recently introduced Music Arena (Kim et al., 2025) provides a live platform for comparative text-to-music evaluation. While these resources represent progress, they primarily focus on text-to-music alignment. **CMI-Pref** fills this gap with large-scale preferences for compositional instructions, including lyrics and audio-to-audio conditioning.

## 3. Method

### 3.1. Data Sources for Training and Benchmarking

We use CMI-Pref-Pseudo for Bradley–Terry pre-training, the training split for CMI-Pref and MusicEval for fine-tuning, and the test split of CMI-Pref, MusicEval along with the full PAM and Music Arena datasets for benchmarking.

**PAM.** We include 500 audio clips from the music subset. Each clip is associated with MOS annotations for musicality and text-music alignment with a text description.

**MusicEval.** We utilize the training and test splits,which provide expert-validated MOS for musicality (musical impression). Due to mismatches among audio files, filenames, and text prompts, we omit its MOS for text-music alignment.

**Music Arena.** We process 2,800 historical interaction logs from the Arena platform up to Dec 2025. To ensure high-quality preference pairs, we remove failed generations and filter out "tie" or "both bad" labels, since users may apply different tolerance margins when two audios are similar. This yields 1,340 clean preference pairs. We categorize Music Arena labels as MUSICALITY; further discussion is provided in Appendix G.1.

We compare additional resources in Table 1, including AIME, MusicPref, and SongEval, but do not include them as main benchmark test sets: MusicPref follows a protocol similar to Music Arena, while Music Arena is more up-to-date and contains more in-the-wild prompts from a live platform; AIME provides two preference dimensions but is limited to text-to-music generation with label-composed prompts; and SongEval does not provide the generation prompts needed for prompt-alignment evaluation. These datasets lack official splits aligned with our protocol, and future work could explore them as additional training sets.

#### 3.1.1. CMI-PREF AND CMI-PREF-PSEUDO: LARGE-SCALE COMPOSITIONAL MULTIMODAL PREFERENCE DATASETS

**CMI-Pref** captures human preferences when music is conditioned on CMIs, including text descriptions, lyrics, and audio prompts. **CMI-Pref-Pseudo** provides large-scale LLM-judge labels for the same setting.

*Table 1.* Statistics of CMI-Pref and CMI-Pref-Pseudo compared with previous datasets. Samples here refer to pairs for preference datasets and individual audio clips for MOS datasets.

| DATA SET | PAM | MUSICEVAL | SONGEVAL | AIME | MUSICPREF | MUSIC ARENA | CMI-PREF-PSEUDO | CMI-PREF |
|---|---|---|---|---|---|---|---|---|
| TEXT CONDITION | ✔ | UNAVAILABLE | ✗ | ✔ | ✔ | ✔ | ✔ | ✔ |
| LYRICS CONDITION | ✗ | ✗ | UNAVAILABLE | ✗ | ✗ | ✔ | ✔ | ✔ |
| AUDIO CONDITION | ✗ | ✗ | ✗ | ✗ | ✗ | ✗ | ✔ | ✔ |
| #SAMPLES | 500 | 2748 | 2,399 | 15,600 | 2,520 | 2,800 | 110K | 4,027 |
| #TESTING SAMPLES | 500 | 413 | - | - | - | 1,340 | - | 500 |
| #DURATION OF AUDIO(HOURS) | 0.83 | 16.62 | 140.54 | 16.67 | UNAVAILABLE | 88.30 | 808.85 | 133.80 |
| #DURATION OF REFERENCE AUDIO(HOURS) | - | - | - | - | - | - | 178.36 | 48.56 |
| #UNIQUE PROMPTS | 100 | 384 | - | 500 | 2,617 | 883 | 10,213 | 2,632 |
| #MODELS & APIS | 5 | 31 | 7 | 12 | 7 | 17 | 23 | 23 |
| OFFICIAL SPLIT | FULL | ✔ | ✗ | ✗ | ✗ | ✗ | - | ✔ |

**Data Collection** We distilled audio from a diverse set of 12 models and 11 commercial APIs to ensure a broad distribution of quality and style. For commercial APIs or products, we generated samples using Suno (v3.5, v4, v4.5, v4.5-plus, v5)[1], Stable Audio 2.0[2], Minimax-Music-2.0[3], Mureka (v7.5, o2)[4], and Loudly[5]. These include equal splits of instrumental and vocal tracks (with lyrics) if the model supports lyrics as input, and equal splits of input with and without audio prompts if applicable.

For open-source models, we generated audio from Music-Gen (Copet et al., 2023), Stable Audio Open (Evans et al., 2025), YUE (Yuan et al., 2026), SongGen (Liu et al., 2025c), AudioLDM (Liu et al., 2023), AudioLDM 2 (Liu et al., 2024), DiffRhythm (Ning et al., 2025), Levo (Lei et al., 2026), Magenta Lyria-RealTime (Team et al., 2025), Jamify (Liu et al., 2025b), MusicLDM(Chen et al., 2024a), and ACE-step (Gong et al., 2025). 35.6% of samples are conditioned on audio prompts for style transfer or continuation in addition to text and lyrics. Audio caption of the audio prompt is provided by Qwen3-Omni as additional text condition if model input cannot support audio prompt.

**Annotation and Statistics** The CMI-Pref-Pseudo dataset initially includes 130k samples, while retaining 110k pairs after consistency checks (See Appendix B), spanning 47,546 generations (797.34h). It is generated by prompting Qwen3-Omni with the prompts in Appendix K, yielding two-dimensional preference pairs for musicality and prompt alignment.

Following the annotation protocol described in Appendix C, 31 annotators constructed CMI-Pref. For each pair, annotators choose preferences for both musicality and prompt alignment. Each vote additionally includes a 1–5 confidence score. A rationale explaining the decision along the two dimensions is also included for future research.

CMI-Pref comprises 4,027 preference samples from generations produced by 23 models, with a total duration of 133.8 hours. We reserve a balanced 500-pair test set, with a 1:1:1:1 split across text, text+lyrics, text+audio, and text+audio+lyrics conditions. Three annotators re-annotated all 500 test pairs; agreement with the original labels reaches 75.2% for musicality and 75.0% for alignment. Detailed train/test agreement is provided in subsection A.3, and modality statistics are provided in Appendix A. Table 1 highlights our distinct advantages in scale, duration, and modality diversity over existing benchmarks.

### 3.2. CMI-RewardBench

We introduce CMI-RewardBench, a unified benchmark designed to evaluate the capability of music reward models in capturing human aesthetic and instructional preferences. It uses only held-out data: **PAM** provides 500 scalar ratings for musicality and text-music alignment; the test split of **MusicEval** provides 413 scalar musicality ratings; **Music Arena** provides 1,340 filtered pairwise preferences; and the test split of **CMI-Pref** provides 500 pairwise musicality and prompt-alignment labels under compositional conditions.

#### 3.2.1. EVALUATION PROTOCOL

To ensure robust evaluation across heterogeneous label formats, we employ two protocols: regression-based correlation and preference-based accuracy. For scalar ratings in PAM and MusicEval, we report Linear Correlation Coefficient (LCC), Spearman Rank Correlation (SRCC), and Kendall-Tau (K-Tau). For pairwise labels in Music Arena and CMI-Pref, models must select the preferred audio, and we report accuracy against human annotations.

#### 3.2.2. BASELINE MODELS

We benchmark a diverse set of current state-of-the-art models, categorized by their primary training objective. The musicality baselines includes specialized MOS predictors such as PAM (Deshmukh et al., 2024), audiobox (Tjandra et al., 2025) and SongEval (Yao et al., 2025). PAM is a reference-free metric leveraging audio-language models for

---

[1]https://suno.com/home
[2]https://platform.stability.ai/
[3]https://platform.minimaxi.com/
[4]https://platform.mureka.ai/
[5]https://www.loudly.com/

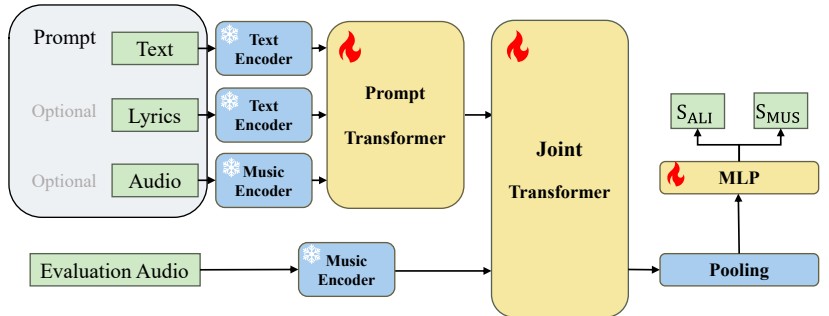

*Figure 2.* Model architecture of compositional music instruction reward model (CMI-RM).

general-purpose quality assessment. Audiobox-Aesthetic is an assessment framework providing four specialized metrics: Production Quality (PQ), Production Complexity (PC), Content Enjoyment (CE), and Content Usefulness (CU). SongEval is a model specifically trained on full-length songs using five aesthetic dimensions: coherence, memorability, naturalness, clarity, and overall musicality.

For text-music alignment, we examine similarity-based metrics: We use CLAP-Score in both default (Wu et al., 2023) and suggested music-specialized checkpoints (Grötschla et al., 2025), as well as CLAMP3 (Wu et al., 2025), a state-of-the-art multimodal alignment model for ABC symbolic notation, waveform and text. MuQ-MuLan(Zhu et al., 2025) is a joint music-text embedding framework utilizing self-supervised music representation learning.

We also include a general-purpose open-source reward model, Omni-Reward (Jin et al., 2025). Furthermore, we evaluate the zero-shot capabilities of frontier AudioLLMs for all three tasks: musicality, text-music alignment, and compositional instruction alignment. The evaluated suite includes the Qwen series and Gemini series: Qwen2-Audio (Chu et al., 2024), Qwen2.5-Omni (Xu et al., 2025a), and Qwen3-Omni, Gemini 2.5 Flash, 2.5 Pro (Comanici et al., 2025), and 3 Pro.

### 3.3. Compositional Music Reward Modeling

We develop a reward model architecture to handle multimodal conditioning and predict fine-grained human scores.

#### 3.3.1. MODEL ARCHITECTURE

**Task Formulation.** Given a compositional prompt

$$\mathcal{P} = (t, l, a_{\text{ref}}),$$

where $t$ denotes optional text description, $l$ optional lyrics, and $a_{\text{ref}}$ optional reference audio, together with an evaluation audio $a_{\text{eval}}$, our reward model predicts human-aligned

preferences over generated music along the two complementary dimensions musicality (MUS) and alignment (ALI). The model outputs two scalar scores $(s_{\text{MUS}}, s_{\text{ALI}}) \in \mathbb{R}^2$.

**Architecture.** We adopt a two-tower multimodal architecture following Zhang et al. (2024b). One tower encodes the multimodal prompt, while the other processes the target audio. All encoders are frozen and instantiated from MuQ-MuLan. Text descriptions $t$ and lyrics $l$ are encoded separately using the text encoder, while reference audio $a_{\text{ref}}$ and evaluation audio $a_{\text{eval}}$ are encoded using the audio encoder. Empty modalities are treated as zero tensors during training and inference. The encoded prompt components are concatenated and fused using a 4-layer Prompt Transformer:

$$\mathbf{h}_{\text{prompt}} = \text{PromptTF}([\mathbf{E}_t; \mathbf{E}_l; \mathbf{E}_{a_{\text{ref}}}]), \tag{1}$$

where $[\cdot; \cdot]$ denotes sequence concatenation.

To model interactions between the prompt and the generated music, the fused prompt embedding and the evaluation audio embedding are concatenated and processed by a single-layer self-attention Joint Transformer:

$$\mathbf{h}_{\text{prompt}}; \mathbf{h}_{\text{eval}} = \text{JointTF}([\mathbf{h}_{\text{prompt}}; \mathbf{E}_{a_{\text{eval}}}]). \tag{2}$$

We extract the hidden states corresponding to the evaluation audio tokens, apply temporal pooling, and project them through a lightweight MLP to obtain the final scores:

$$(s_{\text{ALI}}, s_{\text{MUS}}) = \text{MLP}(\text{Pool}(\mathbf{h}_{\text{eval}})). \tag{3}$$

#### 3.3.2. TRAINING STRATEGY

We train the reward model using a two-stage pipeline that leverages both large-scale pseudo-labeled data and high-quality human annotations. The datasets involved include CMI-Pref-Pseudo for preference pre-training, together with the training split of CMI-Pref and the training split of MusicEval for expert fine-tuning. Both MUS and ALI heads

are optimized jointly throughout both stages. Given a training sample, we compute the musicality-related loss $\mathcal{L}_{\text{MUS}}$ and the alignment-related loss $\mathcal{L}_{\text{ALI}}$ when applicable. The overall training objective is an average of the two:

$$\mathcal{L}_{\text{total}} = \frac{1}{2}(\mathcal{L}_{\text{MUS}} + \mathcal{L}_{\text{ALI}}) \quad (4)$$

**Stage 1: Preference Pre-training.** We first pre-train the model on preference pairs from CMI-Pref-Pseudo. Training is conducted for 2k steps with a batch size of 48. Pairwise preferences are modeled using Bradley–Terry (Bradley & Terry, 1952) formulation. Given a prompt $\mathcal{P}$ and two candidate audio files $A$ and $B$,

$$P(A > B) = \sigma\big(s_\theta(\mathcal{P}, A) - s_\theta(\mathcal{P}, B)\big), \quad (5)$$

where $s_\theta$ denotes the predicted MUS or ALI score depending on the annotation type. The model is optimized using cross-entropy loss, and tied preferences are excluded from training. To mitigate over-confident decision boundaries induced by noisy pseudo labels, we apply label smoothing with a ratio of 0.2 during this stage. Furthur explanation can be found in Appendix E.1.

**Stage 2: Expert Fine-tuning.** We then fine-tune the model on a mixture of high-quality human annotations, combining the training split of CMI-Pref and MusicEval, resulting in a total of 6,647 training samples. We use a batch size of 48 and perform early stopping based on validation performance. The selected checkpoint is obtained after 250 optimization steps selected with early stopping. Human annotations appear in two formats: (1) pairwise preferences $(\mathcal{P}, A, B)$, trained using Bradley–Terry loss as in Stage 1; and (2) scalar ratings $(\mathcal{P}, A, y)$, where $y \in [1, 5]$. For scalar ratings, we regress the predicted scores using

$$\mathcal{L}_{\text{reg}} = \text{MSE}\big(2\tanh(as + b) + 3, \ y\big), \quad (6)$$

where $s$ denotes the predicted MUS or ALI score. The scaling parameters are initialized to $a = 0.2$ and $b = 0$ during fine-tuning. We drop these constants during inference.

### 3.3.3. TEST-TIME SCALING

To evaluate the efficacy of test-time scaling (Jin et al., 2026), we conduct experiments using two backbone models: **MusicGen-small** and **Stable-Audio-Open-small**. For each of the 2,183 text prompts from the MusicCaps (Agostinelli et al., 2023) dataset eval-split, we generate 10 audio samples (10 sec each) per model. Our reward model serves as a "best-of-N" filter to select the top-performing sample, where $N \in \{1, 3, 10\}$. We evaluate the effectiveness via subjective A/B testing: Validating whether reward-model selection consistently aligns with human preferences for superior musical quality. The selection criterion is the average of musicality and alignment scores. Details of the subjective test can be found in Appendix H.2.

## 4. Discussion

### 4.1. Benchmark Results

#### 4.1.1. COMPREHENSIVE EVALUATION ON MUSICALITY

**Superior Generalization on Musicality.** Table 2 presents a quantitative comparison of our proposed **CMI-RM** against all baselines. Our method demonstrates robust generalization capabilities across diverse regression tasks. Specifically, the model fine-tuned on CMI-Pref achieves state-of-the-art performance on the PAM music subset (e.g., SRCC of **0.6988**). Notably, while specialized baselines such as SongEval-RM exhibit strong performance on the Music Arena platform, their efficacy degrades on the more compositionally complex CMI-Pref dataset (72.40%). In contrast, by incorporating external data (w/ f.t.: CMI + MusicEval), our CMI-RM maintains highly competitive accuracy on Music Arena (73.43%) while achieving state-of-the-art preference alignment on CMI-Pref (**78.20%**). This indicates that our preference alignment strategy yields robust representations that consistently correlate with granular human ratings across varying evaluation scenarios.

**Deficiency of General-Purpose MLLMs.** Frontier MLLMs remain weaker for fine-grained music judgment: on CMI-Pref musicality, Gemini 3 Pro and Qwen3-omni obtain 65.80% and 60.40%, versus **78.20%** for CMI-RM. Some AudioLLMs (e.g., Qwen2-audio) often fail the required decision format, leading to near-random results.

#### 4.1.2. COMPOSITIONAL MULTIMODAL INSTRUCTION ALIGNMENT EVALUATION

Table 3 displays the compositional alignment evaluation. Unlike standard benchmarks that primarily focus on text-to-music consistency, CMI-RewardBench utilizes the CMI-Pref dataset to evaluate the capability of models to follow complex instructions involving text, lyrics, and reference audio simultaneously.

**Comparison with Objective Metrics and General MLLMs.** Standard metrics and general MLLMs are limited under compositional conditions. CLAP (default) generally yield moderate performance (about 60–64% on CMI-Pref subsets) but cannot handle composed audio-conditioned cases. Qwen3-Omni is strong on PAM (LCC 0.5841) but weaker on CMI-Pref w/ audio (64.0%). CMI-RM reaches **70.20%** on CMI-Pref w/o audio (CMI+MusicEval) and **79.20%** on w/ audio (CMI-Pref fine-tuned), surpassing Gemini 2.5 Pro (67.2% and 72.8%). This validates that CMI-RM provides a unified solution for diverse and multimodal generation conditions.

**Detailed Breakdown by Modality.** On the hardest Text+Lyrics+Audio subset, CMI-Pref fine-tuning reaches **82.40%**, well above Gemini 3 Pro (66.8%). For w/o-audio

*Table 2.* Musicality results on CMI-RewardBench. For each metric, the best performance is marked in boldface and second with underline.

| Musicality Method&Model | PAM (Music Subset) | | | MusicEval (Test Split) | | | Music Arena ACC | CMI-Pref ACC |
|---|---|---|---|---|---|---|---|---|
| | LCC | SRCC | K-Tau | LCC | SRCC | K-Tau | | |
| PAM score | 0.5873 | 0.6099 | 0.4367 | 0.6466 | 0.6724 | 0.4874 | 63.13% | 65.40% |
| audiobox-CE | 0.5283 | 0.5204 | 0.3665 | 0.6393 | 0.6599 | 0.4830 | 64.25% | 71.80% |
| audiobox-CU | 0.4645 | 0.4704 | 0.3279 | 0.6272 | 0.6764 | 0.4950 | 67.76% | 71.40% |
| audiobox-PC | 0.2505 | 0.2230 | 0.1552 | 0.1225 | 0.0768 | 0.0514 | 58.73% | 59.00% |
| audiobox-PQ | 0.4636 | 0.4513 | 0.3166 | 0.6016 | 0.6335 | 0.4620 | 67.54% | 73.80% |
| SongEval-RM | **0.6987** | 0.6977 | 0.4997 | 0.7140 | 0.6949 | 0.5185 | **73.88%** | 72.40% |
| Omni-Reward | 0.3364 | 0.3115 | 0.2128 | 0.5306 | 0.5137 | 0.3642 | 54.03% | 65.60% |
| Qwen2-audio | 0.1468 | 0.1523 | 0.1120 | 0.1455 | 0.2196 | 0.1585 | 5.99% | 8.60% |
| Qwen2.5-omni | 0.2776 | 0.2837 | 0.2144 | 0.1655 | 0.1454 | 0.1145 | 36.05% | 17.40% |
| Qwen3-omni | 0.4155 | 0.4113 | 0.3146 | 0.3693 | 0.3101 | 0.2205 | 59.63% | 60.40% |
| Gemini2.5-flash | 0.3813 | 0.3693 | 0.2571 | 0.4188 | 0.3886 | 0.2694 | 64.12% | 64.20% |
| Gemini2.5-pro | 0.4463 | 0.4355 | 0.3068 | 0.4966 | 0.4902 | 0.3454 | 69.75% | 70.00% |
| Gemini3-pro | 0.5972 | 0.5967 | 0.4283 | 0.6044 | 0.6018 | 0.4400 | 68.85% | 65.80% |
| - w/o f.t.: Distill only | 0.4358 | 0.4304 | 0.2981 | 0.5253 | 0.5117 | 0.3711 | 64.25% | 70.80% |
| - w/ f.t.: CMI-Pref | 0.6932 | **0.6988** | **0.5101** | 0.7272 | 0.7315 | 0.5495 | 71.41% | 77.80% |
| - w/ f.t.: CMI + MusicEval | 0.6367 | 0.6606 | 0.4754 | **0.8195** | **0.8266** | **0.6459** | 73.43% | **78.20%** |

subsets, CMI+MusicEval gives the best result (**70.20%**), indicating complementary gains from MusicEval.

## 4.2. The Effectiveness of Different Training Sets

We ablate how training data sources affect CMI-RM under a fixed architecture and identical trainable parameters. All variants share the same two-head setup (MUS/ALI) and differ only in (i) initialization and (ii) fine-tuning data.

**Training variants.**

- **Distill only**: Pre-trained on CMI-Pref-Pseudo only (pairwise preference, 1.5 epochs).
- **Distill+CMI-Pref**: Initialized from **Distill**, then finetuned on CMI-Pref (train split).
- **Distill+MusicEval**: Initialized from **Distill**, then finetuned on MusicEval (train split).
- **Distill+Both**: Initialized from **Distill**, then jointly finetuned on CMI-Pref + MusicEval.
- **Scratch+Both**: Random initialization, trained on CMI-Pref + MusicEval.

**Aggregated metric.** To summarize results across heterogeneous benchmarks, we report one aggregated score per dataset in Table 4. For PAM, we average SRCC over the *musicality* and *text–music alignment* regression tasks. For MusicEval, we report SRCC on its musicality MOS. For Music Arena, we report the accuracy of the preference, where the prediction of the model uses the difference between the musicality score $s_{MUS}$ of the pair. For CMI-Pref, we report mean accuracy averaged over musicality and alignment preferences. Detailed per-task numbers are provided in Table 2,

Table 3, and Table 4.

*Table 4.* Aggregated ablation results with a fixed CMI-RM architecture. Mean SRCC on PAM averages the *musicality* and *text–music alignment* regression tasks. Music Arena accuracy uses musicality score $s_{MUS}$ for pairwise prediction. Mean Acc. on CMI-Pref averages musicality and alignment preference accuracy.

| Training variant Metric ↑ | PAM Mean SRCC | MusicEval SRCC | Music Arena Acc. | CMI-Pref Mean Acc. |
|---|---|---|---|---|
| Distill only | 0.3925 | 0.5117 | 64.25% | 71.10% |
| Distill+CMI-Pref | **0.6116** | 0.7315 | 71.41% | 75.90% |
| Distill+MusicEval | 0.4338 | 0.3460 | 64.78% | 69.00% |
| Distill+Both | 0.5464 | **0.8266** | **73.43%** | **76.05%** |
| Scratch+Both | 0.2630 | 0.4986 | 71.34% | 72.15% |

**Findings.** **(1) CMI-Pref is the primary driver for crossbenchmark generalization.** Fine-tuning on CMI-Pref consistently improves all four aggregated metrics: **Distill+CMI** yields consistent improvements over the **Distill** baseline on PAM (0.6116 vs. 0.3925), MusicEval (0.7315 vs. 0.5117), Music Arena (71.41% vs. 64.25%), and CMI-Pref (75.90% vs. 71.10%). This suggests that high-quality human preference data, when coupled with compositional conditions, provides highly transferable supervision signals.

**(2) MusicEval provides a complementary signal, but is insufficient alone.** While **Distill+MusicEval** alone does not yield consistent gains across all sets, **Distill+Both** effectively integrates both datasets. This joint training substantially improves MusicEval correlation (to **0.8266**) while pushing Music Arena and CMI-Pref to their peak performances (**73.43%** and **76.05%**, respectively). We observe a mild trade-off on PAM compared to **Distill+CMI**, indicating a slight objective mismatch between PAM and MusicEval that the joint training mechanism partially mitigates, while

*Table 3.* Benchmark Results on Compositional Multimodal Instruction Alignment.

| | PAM (Music Subset) | | | CMI-Pref (Subsets) | | | | | |
|---|---|---|---|---|---|---|---|---|---|
| Text-Music | ✔ | ✔ | ✔ | ✔ | ✔ | ✔ | ✔ | CMI-Pref w/o Audio | CMI-Pref w/ Audio |
| Lyrics-Music | ✗ | ✗ | ✗ | ✗ | ✔ | ✗ | ✔ | | |
| Audio-Music | ✗ | ✗ | ✗ | ✗ | ✗ | ✔ | ✔ | | |
| Metrics | LCC | SRCC | K-Tau | ACC | ACC | ACC | ACC | ACC | ACC |
| CLAP score (default) | 0.4692 | 0.4517 | 0.3171 | 60.80% | 64.00% | - | - | 62.40% | - |
| CLAP score (music) | 0.3192 | 0.2881 | 0.1978 | 67.20% | 73.60% | - | - | **70.40%** | - |
| MuQ-Mulan | 0.4984 | 0.4741 | 0.3341 | 64.80% | 68.00% | - | - | 66.40% | - |
| CLAMP3 score | 0.2998 | 0.3013 | 0.2068 | 63.20% | 62.40% | - | - | 62.80% | - |
| Omni-Reward | 0.3376 | 0.3072 | 0.2120 | 56.80% | **76.80%** | 67.20% | 70.40% | 66.80% | 68.80% |
| Qwen2-audio | -0.024 | -0.025 | -0.020 | 0.80% | 2.40% | 8.80% | 14.40% | 1.60% | 11.60% |
| Qwen2.5-Omni | 0.1529 | 0.2084 | 0.1696 | 31.20% | 37.60% | 25.60% | 32.00% | 34.40% | 28.80% |
| Qwen3-Omni | **0.5841** | **0.5907** | **0.4714** | 67.20% | 60.00% | 64.80% | 63.20% | 63.60% | 64.00% |
| Gemini2.5-Flash | 0.3686 | 0.2454 | 0.1851 | 65.60% | 56.00% | 69.60% | 54.40% | 60.80% | 62.00% |
| Gemini2.5-Pro | 0.4562 | 0.4179 | 0.3192 | **71.20%** | 63.20% | 73.60% | 72.00% | 67.20% | 72.80% |
| Gemini3-Pro | 0.5201 | 0.5373 | 0.4047 | 67.20% | 60.80% | 68.80% | 64.80% | 64.00% | 66.80% |
| - w/o f.t.: Distill only | 0.3647 | 0.3547 | 0.2449 | 67.20% | 72.00% | 69.20% | 77.20% | 69.60% | 73.20% |
| - w/ f.t.: CMI-Pref | 0.5200 | 0.5243 | 0.3721 | 64.80% | 72.80% | 76.00% | **82.40%** | 68.80% | **79.20%** |
| - w/ f.t.: CMI + MusicEval | 0.4418 | 0.4321 | 0.3008 | 67.60% | 72.80% | **76.40%** | 79.20% | 70.20% | 77.80% |

more diverse data is required for better generalization.

**(3) Distillation initialization is universally beneficial.** Comparing **Distill+Both** with **Scratch+Both**, we observe that distillation initialization yields consistent gains across all benchmarks. Most notably, it improves performance on Music Arena from 71.34% to **73.43%**, and significantly boosts both PAM (0.2630 to 0.5464) and CMI-Pref (72.15% to **76.05%**). These results suggest that our large-scale pseudo-label pre-training (**Distill**) establishes a robust prior that not only mitigates overfitting but also transfers effectively to noisy, out-of-distribution scenarios such as Music Arena, thereby establishing **Distill+Both** as the optimal training strategy.

**(4) Independence on Qwen3-Omni pseudo labels.** Our RM is not simply copying the pseudo-labeler. On 500 CMI-Pref-test samples, The agreement rate for two inferences of Qwen-Omni3 is 94.4% musicality / 94.1% alignment, but its agreement with our pseudo-pretrained model after label smoothing is only 80.6% / 79.2%, and drops to 63.7% / 68.4% after human finetuning. This shows human supervision shifts the decision boundary away from the teacher. Distill+CMI-Pref and Distill+Both also outperform Qwen3-Omni on MusicArena, improving accuracy from 59.63% to 71.41% with 1,340 human labels. Adding a second pseudo source (40k Gemini samples) drops the model agreement of the pretrained checkpoint with Qwen3-Omni to 65.2% and 67.3%, but yields only minor final gains shown in Table 14.

### 4.3. Test-Time Scaling

Table 5 shows that RM-based best-of-$N$ reranking provides consistent test-time scaling gains across both backbones. The improvements are more pronounced for **MusicGen-small**, where MUQ-MULAN and AudioBox/SongEval metrics increase monotonically from $N=1$ to $N=10$. For **Stable-Audio-Open-small**, the gains are smaller and begin to saturate, suggesting diminishing returns when candidate quality is concentrated.

The human preference matrices in Fig. 3 provide two takeaways. First, **Ground Truth (GT) is consistently preferred over all reranked outputs** for both backbones, indicating that reranking improves generation quality but does not close the gap to real data. Second, preferences among Top-$k$ selections are *not strictly monotonic* with $N$, and the gains can saturate (e.g., Stable Audio shows limited differences between $N=3$ and $N=10$). Overall, best-of-$N$ reranking is a simple and effective test-time scaling strategy, with diminishing returns beyond moderate $N$.

## 5. Conclusion

We introduce **CMI-RewardBench**, a unified benchmark for reward modeling under compositional multimodal instruction (text, lyrics, and reference audio), together with two datasets: **CMI-Pref-Pseudo** (110k pseudo-labeled pairs) and **CMI-Pref** (4k expert annotations with confidence).

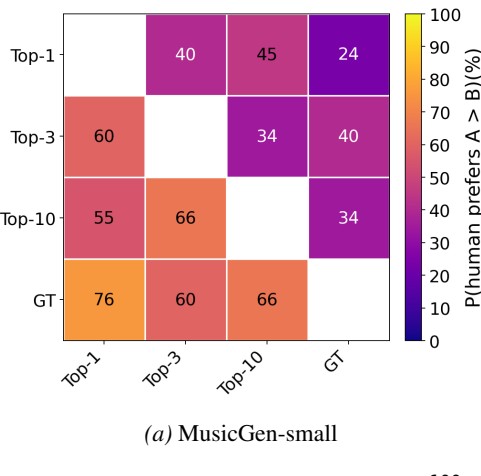

*(a)* MusicGen-small

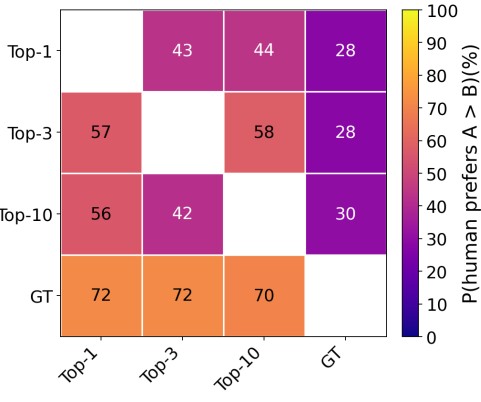

*(b)* Stable-Audio-Open-small

*Figure 3.* Pairwise preference matrices for test-time scaling with RM reranking. Each cell reports the percentage of trials in which annotators preferred system $A$ (row) over system $B$ (column).

*Table 5.* Test-time scaling results of MusicGen-small and Stable-audio-open-small on objective metrics on text-music alignment (MuQ-Mulan) and musicality (AudioBox, and SongEval).

| Model | Alignment MuQ-MuLan↑ | AudioBox CE↑ | AudioBox CU↑ | SongEval Mus↑ |
|---|---|---|---|---|
| MusicGen | 0.298 | 6.046 | 6.989 | 2.143 |
| MusicGen (N=3) | 0.323 | 6.405 | 7.255 | 2.213 |
| MusicGen (N=10) | 0.339 | 6.647 | 7.416 | 2.273 |
| Stable Audio | 0.293 | 5.567 | 7.170 | 2.055 |
| Stable Audio (N=3) | 0.301 | 5.732 | 7.245 | 2.078 |
| Stable Audio (N=10) | 0.307 | 5.799 | 7.290 | 2.090 |

tion of the Gemini series, a family of models developed by Google, this support is disclosed as a potential financial conflict of interest. The authors declare no other financial conflicts of interest.

## Acknowledgement

Yinghao Ma is a research student at the UKRI Centre for Doctoral Training in Artificial Intelligence and Music, supported by UK Research and Innovation [grant number EP/S022694/1]. Yinghao Ma also acknowledges the support of Google PhD Fellowship. Hewei Gao is a PhD student partially supported by alignAI, the alignAI Doctoral Network, funded by the European Union's Marie Skłodowska-Curie Actions programme [grant number 101169473]. We thank Termeh Taheri on suggestions on back-end development and Christos Plachouras on feedbacks on platform design.

## Impact Statement

This work advances the evaluation and alignment of music generation systems by providing a unified benchmark and lightweight reward models that better reflect human preference under compositional multimodal conditions. Our pipeline includes audio generated via commercial APIs; therefore we follow a TOS-aware release policy: only components that are explicitly redistributable are made public, and any restricted parts are shared (if at all) via an application-based access process. We also acknowledge potential copyright concerns from style/melody similarity in generated music and provide a takedown/correction mechanism. Human preference data are collected via a Music-Arena-style platform under informed consent and data minimization (no personal identifiers; limited metadata), with an opt-out/withdrawal mechanism. Given previous datasets included in benchmark are all CC-BY or CC-NC license, our datasets and benchmark are released under **CC-BY-NC-SA** license and accompanied by a datasheet/data card documenting sources, licenses/TOS compatibility, and the public release fields.

By integrating CMI-Pref with existing resources (e.g., PAM, MusicEval, Music Arena), CMI-RewardBench provides a rigorous testbed spanning five tasks from absolute scoring to pairwise preference, revealing a clear capability gap that even frontier multimodal LLM LLM judges struggle to reach strong agreement with expert preferences in this setting.

We further developed **CMI-RM**, a parameter-efficient reward model that supports compositional conditioning over text, lyrics, and audio within a single architecture, achieving performance competitive with or exceeding specialized open-source baselines, and providing measurable gains when used for best-of-$N$ reranking as a simple inference-time scaling strategy. We release the dataset, benchmark, and model weights to facilitate future research.

## Conflict of Interest Disclosure

Yinghao Ma also acknowledges support from the Google PhD Fellowship. Because this work includes an evalua-

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

# A. Detailed Metadata of Datasets

## A.1. Diversity of Prompts

On the basis of a text prompt, the retained CMI-Pref-Pseudo split covers audio+lyrics (18.3%), audio-only (17.0%), lyrics-only (19.8%), and text-only (44.8%) conditions. CMI-Pref contains audio+lyrics (15.2%), audio-only (20.4%), lyrics-only (14.9%), and text-only (49.5%) conditions, with the 500-pair test split balanced across the four prompt modalities (125 pairs each).

CMI-Pref contains a highly diverse collection of prompts and lyrics that reflect realistic and heterogeneous user instructions for music generation. Across the full dataset, we collect 10,213 unique prompts, while the human-annotated split contains 2,788 unique prompts. Prompt lengths follow a long-tailed distribution: most prompts are concise style or intent descriptors consisting of a few words, while a non-trivial subset includes longer compositional instructions specifying multiple musical attributes. This range captures both minimal user inputs and more detailed requests that require structured reasoning from reward models.

Semantically, the prompts span a wide variety of musical attributes. They cover a broad spectrum of genres, including popular, electronic, rock, jazz, classical, ambient, folk, and orchestral styles, with many prompts combining multiple genre cues. Beyond genre, prompts frequently specify mood, tempo, instrumentation, and production characteristics, introducing orthogonal dimensions of variation. Such compositional prompts prevent reward models from relying on single-keyword correlations and instead encourage holistic assessment of instruction adherence.

The prompt distribution is also linguistically diverse. While English prompts dominate, the dataset includes non-English and mixed-language prompts, reducing reliance on a single linguistic prior and better reflecting global usage patterns. Importantly, prompts are paired with generations produced by a wide range of music generation models and commercial APIs, ensuring that instruction diversity is not confounded with a specific synthesis pipeline.

In addition to text prompts, CMI-Pref includes lyric-conditioned instructions at a meaningful scale. The human-annotated split contains 840 examples with non-empty lyrics, and the full dataset contains 3,896 such examples. Lyrics vary substantially in structure and length, ranging from short repetitive hooks to multi-stanza verses with clear narrative progression. They include both vocal-focused instructions and lyrics intended to be adapted to different musical styles, posing additional alignment challenges beyond text-to-music generation.

Overall, the diversity of prompts and lyrics in CMI-Pref provides a realistic and challenging testbed for reward mod-

eling. By combining short and long instructions, multiple semantic control dimensions, diverse musical styles, and lyric-conditioned inputs, the dataset supports robust evaluation of reward models under compositional and multimodal instruction settings.

## A.2. Audio Duration and Listening-Time Statistics

To better characterize both the temporal profile of the generated samples and annotator behavior, we analyze three complementary aspects: (i) the distribution of audio duration, (ii) the distribution of total listening time, and (iii) the correlation between these two quantities.

### A.2.1. DISTRIBUTION OF AUDIO DURATION

Figure 4 reports the duration distribution of unique generated audios ($N = 6458$). The distribution is multimodal: while the dominant mass is concentrated in the short-duration region (especially around 30 seconds), additional peaks appear at longer fixed durations. This pattern is consistent with heterogeneous generation constraints across source models, where many earlier systems primarily support short outputs while others use longer default lengths. Overall, the mean (69.4 seconds) remains notably higher than the median (30.0 seconds), indicating a long right tail.

For readability, we cap the displayed histogram height at 500 counts and annotate overflow bins in red (e.g., 1030, 1436, and 2301). In addition, a very small number of samples with duration above 300 seconds are omitted from visualization; the plotted subset still covers more than 99.5% of all samples.

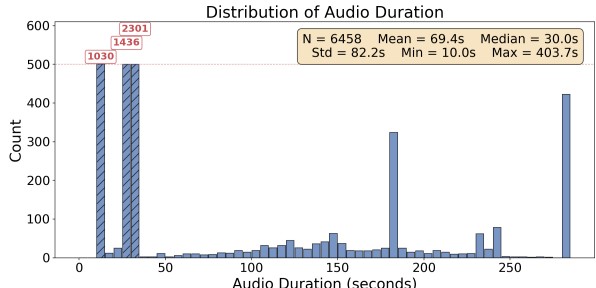

*Figure 4.* Distribution of audio duration. The x-axis denotes generated audio duration (seconds), and the y-axis denotes sample count ($N = 6458$ unique audios). To improve readability, bar heights are clipped at 500 and overflow counts are annotated in red; a tiny fraction of samples with duration >300 seconds is omitted from the plot.

### A.2.2. DISTRIBUTION OF TOTAL LISTENING TIME

We define *total listening time* as the accumulated playback time an annotator spends on one audio within a comparison task. Since CMI-Pref contains 4,027 pairwise votes and

each vote includes two audios, we obtain 8,054 audio-level listening-time records. We exclude 8 extreme records with listening time above 500 seconds (likely idle-page artifacts), resulting in 8,046 valid samples for analysis.

As shown in Figure 5, listening time is also long-tailed, with mean 22.8 seconds and median 13.5 seconds. The majority of samples fall within 50 seconds, and most are above 10 seconds. This pattern suggests that annotators typically listen long enough to form meaningful pairwise judgments, while keeping the annotation process efficient.

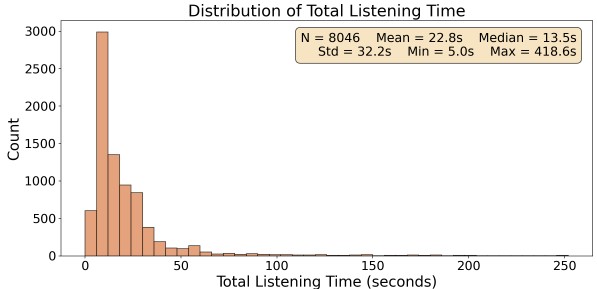

*Figure 5.* Distribution of total listening time per audio. "Total listening time" denotes accumulated playback time spent by annotators on one audio. Starting from 8,054 records (4,027 votes × 2 audios), we remove 8 outliers with listening time >500 seconds and report the remaining $N = 8046$ samples.

### A.2.3. CORRELATION BETWEEN AUDIO DURATION AND TOTAL LISTENING TIME

Figure 6 examines the relationship between audio duration and total listening time on the same filtered set ($N = 8046$). We observe a mild positive correlation (Pearson $r = 0.279$), indicating that longer audios are associated with longer listening on average, but the dependence is limited.

Notably, most points lie below the diagonal $y = x$, meaning annotators often make decisions before fully listening to the entire clip. Meanwhile, the broad spread at fixed duration shows substantial variability in listening behavior across examples. Together, these findings indicate that annotators adaptively allocate listening effort by sample difficulty rather than simply following clip length.

### A.3. Annotator Agreement in CMI-Pref

*Table 6.* Inter-annotator agreement on overlapping votes, computed over 644 vote pairs from 492 comparisons with multiple annotations.

| Metric | Instruction Following | Music Quality |
|---|---|---|
| Agreement Count | 445 | 466 |
| Disagreement Count | 199 | 178 |
| Agreement Rate | 0.691 | 0.724 |
| Krippendorff's $\alpha$ | 0.382 | 0.447 |

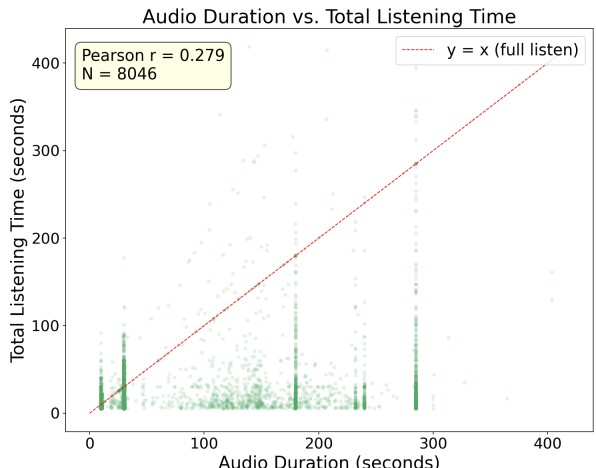

*Figure 6.* Correlation between audio duration (x-axis) and total listening time (y-axis), computed on $N = 8046$ filtered samples. The dashed diagonal denotes $y = x$ (full listening).

*Table 7.* Reliability of CMI-Pref-test, computed by comparing three annotators' re-annotations of all 500 test samples with the original test labels.

| Metric | Alignment Preference | Musicality Preference |
|---|---|---|
| Agreement Rate | 0.750 | 0.752 |
| Krippendorff's $\alpha$ | 0.500 | 0.504 |
| Wilson 95% CI | (70.9%, 78.5%) | (71.1%, 78.7%) |

We analyze inter-annotator consistency for comparisons that received multiple independent votes. A total of 492 comparison pairs were annotated by two or more annotators, producing 1,056 votes. For each comparison, we form all pairwise combinations of its votes, resulting in 644 vote pairs.

We measure agreement using *agreement rate* and *Krippendorff's alpha*, computed separately for alignment and musicality. For the training split, Table 6 reports duplicated-vote agreement on overlapping annotations. Music quality judgments exhibit slightly higher consistency than instruction following, as they are more intuitive and place lower demands on specialized musical knowledge. After correcting for chance agreement, Krippendorff's alpha indicates a *moderate* level of true agreement, which we consider reasonable given the inherent subjectivity of music evaluation.

To directly assess the reliability of CMI-Pref-test, we additionally re-annotated all 500 test samples with 4 annotators. As shown in Table 7, agreement between the re-annotations and the original test labels is substantially higher than the overlap statistics in Table 6. For musicality preference, we obtain $\alpha = 0.504$ and agreement $= 75.2\%$; for alignment preference, we obtain $\alpha = 0.500$ and agreement $= 75.0\%$. We also examined whether model performance is unrealistically high relative to human consistency. The strongest

models are generally close to this re-annotation agreement level: CMI-RM reaches 74% accuracy for alignment and 78% for musicality, while SongEval-RM reaches 72.4% for musicality. This supports the validity of the benchmark rather than suggesting overfitting to noisy labels.

### A.4. Confidence Scores of Human Annotation

We analyze the distribution of annotators' confidence scores (Fig. 7). Overall, most votes are associated with relatively high confidence. The average confidence for music quality is slightly higher than that for instruction following, consistent with earlier observations that music quality judgments are more intuitive.

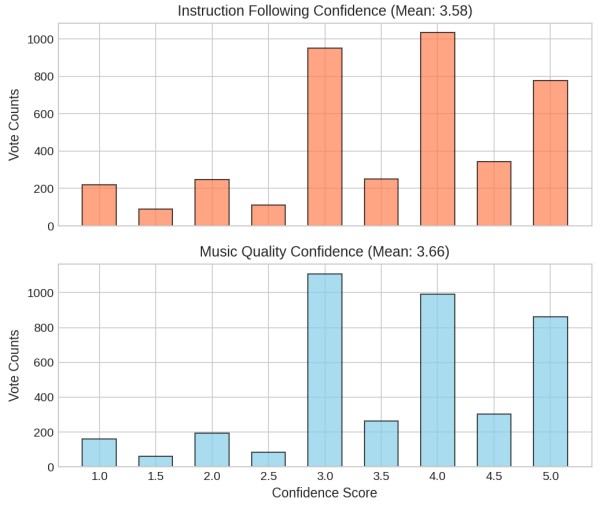

*Figure 7.* Distribution of annotators' confidence scores for instruction following and music quality.

*Table 8.* Agreement between instruction-following and music-quality preferences, and the corresponding average confidence.

| Metric | Value |
|---|---|
| #Votes with same preference | 3309 |
| #Votes with different preference | 718 |
| Agreement rate | 0.822 |
| Avg. confidence (same Pref.) | IF: 3.674, MQ: 3.734 |
| Avg. confidence (diff. Pref.) | IF: 3.171, MQ: 3.311 |

We further examine the relationship between agreement across instruction following and music quality and annotator confidence (Table 8). Overall, the two dimensions show a high agreement rate, and votes have higher average confidence when they agree. This suggests that disagreements between instruction following and music quality often correspond to more ambiguous or trade-off cases, where annotators are less certain about the overall preference.

**Alignment among heads:** We observed a high consistency between musicality and instruction-music alignment pref-

erences. In pseudo-labels, the agreement is 91% between musicality and instruction alignment. In human data, this agreement is lower (81%), but conflicts typically occur when annotators report high confidence in instruction alignment despite lower musicality, highlighting the necessity of evaluating these dimensions separately.

## B. Pseudo-label Generation for CMI-Pseudo

### B.1. Label Acquisition Protocol

To ensure the reliability of our pseudo-labels, we address the well-documented phenomenon of positional bias in Large Language Models (LLMs), where the model's preference is influenced by the presentation order of the options rather than the content quality alone.

Our protocol for collecting pseudo-labels adopts a *Position-Consistency* strategy. For a given pair of audio samples $(A, B)$ generated from the same prompt $P$, we conduct a bidirectional assessment:

1. **Forward Pass:** We query the model with the sequence $(A, B)$ to obtain preference $L_{fwd}$.

2. **Reverse Pass:** We swap the positions to $(B, A)$ and query the model again to obtain preference $L_{rev}$.

A pseudo-label is considered valid and retained only if the judgment is invariant to position—that is, the model prefers the same underlying audio clip in both the forward and reverse passes ($L_{fwd} = L_{rev}$). Comparisons yielding conflicting results or inconsistent ties are discarded as hallucinations or uncertain boundaries.

### B.2. Bias Analysis and Dataset Statistics

We initially sampled 129,545 pairwise comparisons from our generated audio pool. As demonstrated in Table 9, we observe significant distributional shifts when the presentation order is swapped, confirming the presence of positional bias. For instance, in the Musicality metric, the win rate for Candidate A fluctuates from 51.96% in the original configuration to 59.27% in the reversed configuration.

By applying our consistency filter, we retain only the high-confidence labels, resulting in 114,694 valid musicality labels and 117,828 valid alignment labels. As shown in the "Agreed" rows of Table 9, the resulting distribution stabilizes, effectively mitigating the variance introduced by the model's sensitivity to input order. For the final dataset construction, we retained the **intersection** of these valid subsets to ensure quality across all dimensions, yielding approximately 110k pairs.

The "Original" and "Reversed" distributions should theoretically be identical if the evaluator were perfectly unbiased.

*Table 9.* Distribution of pseudo-labels across varying query configurations. "Original" and "Reversed" denote the presentation order of the audio pair. "Agreed" represents the subset of labels where the model's preference remained consistent across both permutations. The discrepancy between Original and Reversed highlights positional bias, which is rectified in the Agreed set.

| Configuration | Win A (%) | Win B (%) | Tie (%) |
|---|---|---|---|
| *Musicality* | | | |
| Original $(A, B)$ | 51.96 | 40.96 | 7.08 |
| Reversed $(B, A)$ | 59.27 | 33.48 | 7.25 |
| **Agreed (Filtered)** | **57.80** | **36.97** | **5.22** |
| *Alignment* | | | |
| Original $(A, B)$ | 33.65 | 43.71 | 22.64 |
| Reversed $(B, A)$ | 38.75 | 38.84 | 22.77 |
| **Agreed (Filtered)** | **36.05** | **41.23** | **22.73** |

The observed divergence necessitates our rigorous filtering approach. We did not shuffle the comparison pairs during analysis, therefore its normal that the labels A, B are uneven.

# C. Human Annotation Details

## C.1. Annotation Protocol for CMI-Pref

### C.1.1. CORE OBJECTIVES

The annotation process is decomposed into three distinct components to ensure a multi-dimensional evaluation of the generated music:

1. **Preference Label (A/B):** A forced-choice selection between two candidates.

2. **Confidence Score (1–5):** A quantitative measure of the annotator's certainty, grounded in constraint satisfaction (for alignment) or quality delta (for musicality).

3. **Free-text Feedback:** Qualitative justifications focusing on fine-grained details that discrete labels cannot capture.

### C.1.2. GENERAL PRINCIPLES

- **Instruction-First:** Annotators must strictly evaluate the *instruction/prompt* before listening to avoid post-hoc rationalization.

- **Holistic and Granular Review:** Each sample is evaluated for overall coherence as well as specific details (instrumentation, emotion, structure, and audio fidelity).

- **Dimensional Isolation:** Annotators are instructed to decouple **Alignment** (adherence to the prompt) from **Musicality** (aesthetic quality and production value). A sample may win in alignment while losing in musicality.

### C.1.3. Q1: TEXTUAL MUSIC ALIGNMENT PREFERENCE

Annotators identify which sample better follows the elements specified in the instruction, regardless of aesthetic appeal. Key alignment dimensions include:

- **Instrumentation:** Specific instruments (e.g., piano solo, guitar riff).

- **Mood/Atmosphere:** Emotional valence (e.g., melancholy, upbeat, tense).

- **Genre/Style/Era:** Stylistic markers (e.g., Lo-fi, Baroque, 80s synth).

- **Rhythm/Tempo:** Temporal characteristics (e.g., driving drum beat, groovy, energetic).

**Confidence Calibration (1–5):**

- **5 (Very Certain):** Clear binary distinction; one satisfies all key constraints while the other fails or collapses.

- **3 (Moderate):** Default choice; a perceptible lean toward one candidate without overwhelming dominance.

- **1 (Uncertain):** Highly ambiguous prompts or both samples are indistinguishable in their failure/success.

### C.1.4. Q2: MUSICALITY PREFERENCE

Annotators evaluate which sample sounds more like a "finished, natural, and professional" musical work, independent of the prompt.

- **Key Criteria:** Melodic memorability, structural progression, rhythmic stability, and production clarity (lack of distortion/artifacts).

- **Confidence:** Reflects the perceived "quality gap" between candidates.

### C.1.5. Q3: FEEDBACK GUIDELINES

Feedback should consist of 1–3 concise sentences focusing on "audible evidence." Annotators are encouraged to use specific timestamps and avoid vague descriptors.

- **Positive Justification:** "Sample A aligns better due to the presence of the specified saxophone; the melody is more distinct."

- **Negative Evidence:** "Sample B suffers from rhythmic instability at 0:20 and harsh high-frequency distortion."

*Table 10.* Taxonomy of musical attributes used in the annotation process.

| Dimension | Positive Descriptors | Negative Descriptors |
|---|---|---|
| Melody | Catchy, Memorable, Distinct | Repetitive, Generic, Wandering |
| Structure | Coherent, Progression, Build-up | Disjointed, Abrupt, Random Loops |
| Rhythm | Groovy, Steady, Driving | Off-beat, Unstable, Chaotic |
| Audio Quality | Clean mix, Balanced, High-fidelity | Harsh, Muddy, Distorted/Clipping |
| Vocals | Natural, Clear articulation | Robotic, Slurred, Artifacts |

### C.1.6. GLOSSARY FOR ANNOTATORS

## C.2. Annotation Platform

Our annotation platform is illustrated in Figure 8. It provides a unified interface for pairwise audio comparison, confidence scoring, and free-text feedback collection. Detailed usage instructions are provided in the README included in the submitted supplementary materials.

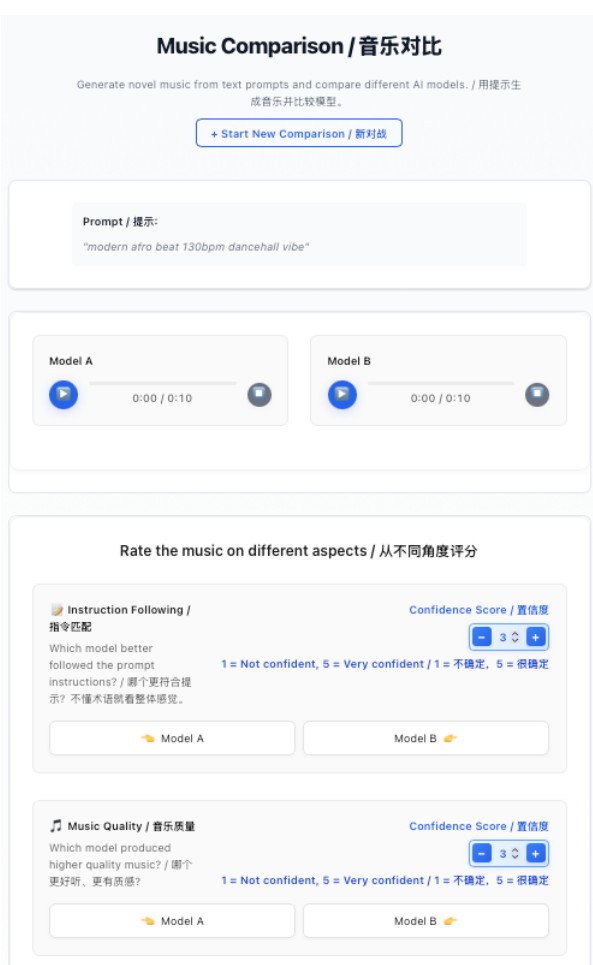

*Figure 8.* Platform of human annotation.

## D. Limitations

Our reward model is designed as a practical **baseline evaluator** for compositional music preference modeling, rather than a complete solution for reward optimization in downstream RL. In particular, we do not claim that a single reward model prevents reward hacking by itself.

We observe **over-correlation between heads**. In human labels, musicality and alignment preferences agree on around **82%** of votes; in model predictions, the SRCC between alignment and musicality scores is around **0.853**. This correlation helps transfer, but also suggests incomplete disentanglement when musicality and alignment conflict.

We also observe **bias by genre and language**. Musicality scores vary more across genres than alignment scores, with highly structured acoustic genres (classical, jazz, pop) tending to receive higher scores than electronic or experimental genres. However, on pairwise preference evaluation, this genre effect is less severe, since lower-scoring genres still show near-average accuracy. We also observe language imbalance: alignment performance is stronger on English than on low-resource languages.

## E. Ablation Study on Pseudo-Labeled Data

### E.1. Distribution Shift and the Effectiveness of Label Smoothing

We first pretrain the reward head on pseudo-labeled data until convergence ($\sim$14k steps), and then fine-tune it on real preference data. Although pretraining on AI feedback already yields reasonably high accuracy on the real test set (**71.0%** on Musicality and **71.8%** on Alignment), it provides only *limited* additional benefit after downstream fine-tuning: pseudo-pretraining without smoothing reaches **75.2%/71.0%** after fine-tuning, comparable to directly fine-tuning from scratch (**75.0%/69.3%**).

We attribute this to a distribution shift between pseudo-labeled and real data, which manifests as over-confident probabilities. Specifically, we track the cross-entropy loss on the real test set, We measure binary cross-entropy (CE)

$$\mathcal{L}_{\text{CE}} = -\big(y \log p_\theta(A \succ B) + (1-y)\log(1-p_\theta(A \succ B))\big),$$

and observe that it can exceed **1.2** initially, while a random guess baseline yields $-\ln(0.5) = 0.693$ for a binary task. This indicates that the model can be confidently wrong under the shift, which is penalized heavily by cross-entropy even when accuracy changes only slightly. Figure 9 shows that without label smoothing, the CE loss on the pseudo validation set remains stable, whereas the CE loss evaluated on real data increases substantially. In contrast, the corresponding accuracy curves in Figure 10 show much smaller differences, suggesting the model produces over-confident

*Table 11.* Real-test reward-head accuracy (%) under different training pipelines.

| Training pipeline | Musicality Acc. | Alignment Acc. |
|---|---|---|
| Pseudo pretrain (before FT) | 71.0% | 71.8% |
| Pseudo pretrain → FT (no smoothing) | 75.2% | 71.0% |
| Direct FT (from scratch) | 75.0% | 69.3% |
| Pseudo pretrain → FT (with smoothing) | **77.8%** | **74.0%** |

decision boundaries.

To mitigate this over-confidence issue, during pseudo training, we apply label smoothing with parameter $\varepsilon$:

$$\tilde{y} = (1 - \varepsilon)y + \varepsilon/2,$$

which replaces hard targets $y \in \{0, 1\}$ with softened targets $\tilde{y} \in (0, 1)$. We also only train the model 1.5 epochs. In our experiments, we set $\varepsilon = 0.2$, i.e., $(1, 0) \to (0.9, 0.1)$. This simple change yields clear downstream gains: pseudo-pretraining *with* smoothing achieves **77.8%** on Musicality and **74.0%** on Alignment after fine-tuning, outperforming both direct fine-tuning and pseudo-pretraining without smoothing.

Label smoothing makes checkpoint selection more robust. With smoothing, CMI-PREF-test accuracy stays around $75.3 \pm 1\%$ from 2k to 20k steps, while training without smoothing is more sensitive to stopping points and yields worse downstream transfer. However, this robustness is not uniform across datasets: Music Arena starts declining after around 6k steps, and PAM is more sensitive to over-training. This is possible due to dataset-specific inconsistencies. Therefore, we still maintain a relative early stopping point (1.5 epochs) to prevent overfitting to specific patterns.

### E.2. Ablation on Pseudo-Label Data Size

We ablate the amount of pseudo-labeled data used for pre-training. To simplify the evaluation protocol, we fine-tune only on CMI-PREF. Unless otherwise noted, label smoothing is enabled during pseudo pre-training.

**Metrics.** We report two groups of metrics. **Transfer before fine-tuning:** we evaluate the pretrained model on the CMI-PREF test set and report (i) **Pref-Test Acc.**, defined as the average reward-head accuracy over the Musicality and Alignment heads, and (ii) **Pref-Test CE**, defined as the corresponding average cross-entropy. **Downstream after fine-tuning:** after fine-tuning on CMI-PREF, we report the head accuracies: **Mus-Acc** (Musicality head accuracy) and **Align-Acc** (Alignment head accuracy).

**Setup.** The full pseudo-labeled dataset contains **110k** examples. We vary the pseudo data size in $\{4k, 8k, 16k, 32k, 64k, 110k\}$. We primarily compare models at a fixed **epoch progress** during pseudo pre-training (Table 12), which provides a natural normalization across

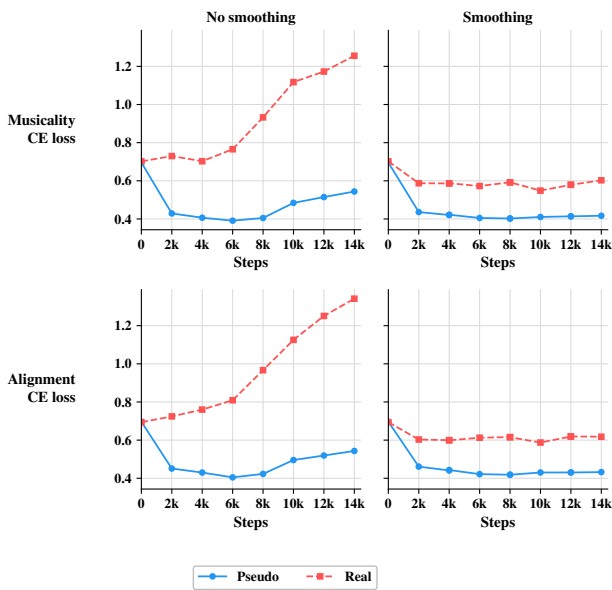

*Figure 9.* Cross-entropy loss of checkpoints trained on CMI-Pseudo. *Pseudo* indicates metrics on the CMI-Pseudo validation set, and *Real* denotes results on the test set of CMI-Pref.

*Table 12.* Pseudo data size ablation under the fixed epoch-progress protocol (same-epoch comparison).

| Pseudo Size | Pref-Test Acc.↑ | Pref-Test CE↓ | Mus-Acc↑ | Align-Acc↑ |
|---|---|---|---|---|
| 4k | 0.582 | 0.674 | 0.726 | 0.664 |
| 8k | 0.640 | 0.656 | 0.750 | 0.692 |
| 16k | 0.605 | 0.655 | 0.754 | 0.700 |
| 32k | 0.646 | 0.627 | 0.782 | 0.700 |
| 64k | 0.711 | 0.593 | 0.778 | 0.740 |
| 110k | 0.720 | 0.559 | 0.761 | 0.735 |

different dataset sizes. For completeness, we also include a fixed-step comparison in Table 13.

**Results.** Under the same-epoch protocol (Table 12), increasing the pseudo data size generally improves both transfer (Pref-Test Acc/CE) and downstream performance. Moving from small pseudo sets (4k–16k) to **32k–64k** yields noticeably stronger results, while performance saturates from **64k** to **110k**. Notably, **64k** achieves performance comparable to using the full pseudo-labeled set under the same-epoch protocol, suggesting diminishing returns beyond this scale. Therefore, we adopt **64k** pseudo-labeled examples as a compute-efficient operating point in subsequent experiments. Since directly finetuning on the 3.5k CMI-Pref yields Pref-Text Acc of 0.7215, we validate our rule-of-thumb that Psuedo data should be more than 10 times compared to real data. The ablation on epoch vs step also demonstrates that epoch is a more natural normalization in this finetuning setting.

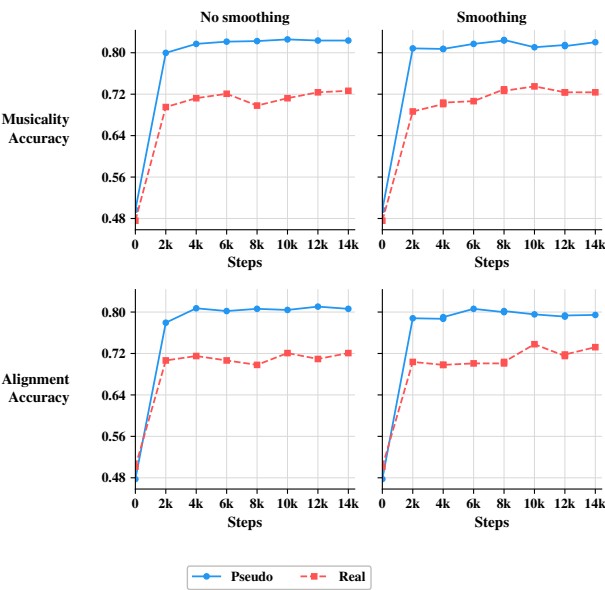

*Figure 10.* Accuracy of checkpoints trained on CMI-Pseudo. *Pseudo* indicates metrics on the CMI-Pseudo validation set, and *Real* denotes results on the test set of CMI-Pref.

*Table 14.* Aggregated results under different pseudo-label source designs.

| Pseudo-label setup | PAM Mean SRCC | MusicEval SRCC | Music Arena Acc | CMI-Pref Mean Acc |
|---|---|---|---|---|
| Smooth (Qwen only) | 0.529 | 0.742 | 0.701 | 0.756 |
| Mix (70k Qwen + 40k Gemini) | 0.550 | 0.761 | 0.699 | 0.768 |
| Smooth + Mix | 0.450 | 0.765 | 0.707 | 0.775 |

*Table 13.* Pseudo data size ablation under the fixed-step protocol (same-step comparison).

| Pseudo Size | Pref-Test Acc.↑ | Pref-Test CE↓ | Mus-Acc↑ | Align-Acc↑ |
|---|---|---|---|---|
| 4k | 0.588 | 0.724 | 0.742 | 0.696 |
| 8k | 0.623 | 0.685 | 0.768 | 0.694 |
| 16k | 0.646 | 0.636 | 0.760 | 0.682 |
| 32k | 0.566 | 0.725 | 0.752 | 0.710 |
| 64k | 0.711 | 0.593 | 0.778 | 0.740 |
| 110k | 0.712 | 0.570 | 0.762 | 0.714 |

### E.3. Ablation on Mixed Pseudo-Label Sources

We also test whether mixing pseudo labels from multiple teachers is more effective than Qwen-only supervision with label smoothing. The mixed setting uses **70k** Qwen3-Omni labels and **40k** Gemini labels. Table 14 shows that source mixing provides only modest and inconsistent gains. Mixing alone improves PAM and CMI-PREF, but slightly hurts Music Arena. Combining smoothing and mixing gives the best CMI-PREF and Music Arena results, but degrades PAM. Overall, label smoothing remains the more reliable regularizer, while multi-source distillation mainly changes the teacher boundary.

*Table 15.* Model size scaling of the trainable reward head.

| Variant | Trainable Params | PAM Mean SRCC | MusicEval SRCC | MusicArena Acc | CMI-Pref Mean Acc |
|---|---|---|---|---|---|
| Small | 8.3M | 0.540 | 0.816 | 0.680 | 0.744 |
| Base | 38M | 0.578 | 0.811 | 0.729 | 0.746 |
| Large | 102M | 0.589 | 0.848 | 0.719 | 0.765 |

*Table 16.* Accuracy change from replacing the default text encoder with Flan-T5-Large. T/L/A denote text, lyrics, and audio conditions.

| Metric | Arena T | Arena T+L | CMI T | CMI T+L | CMI T+L+A | CMI T+A |
|---|---|---|---|---|---|---|
| Δ Acc. | +2.7% | -1.4% | -4.0% | 0.0% | -7.2% | -6.8% |

## F. Reward Model

### F.1. Model Size Scaling

Our main configuration keeps pretrained encoders frozen and trains only the reward head. To study parameter efficiency, we vary hidden size, attention heads, and Transformer depth to obtain three reward-head sizes: **small** (8.3M), **base** (38M), and **large** (102M). All variants are pretrained on CMI-PREF-PSEUDO and then fine-tuned on CMI-PREF and MusicEval.

Table 15 shows that smaller models can still work well: the 8.3M variant remains competitive on PAM, MusicEval, and CMI-PREF, and mainly underperforms on Music Arena. The 102M variant achieves the strongest overall results, especially on MusicEval and CMI-PREF, indicating better multi-task transfer with larger heads.

### F.2. Text Encoder Ablation with Flan-T5

We replace the default text encoder with **Flan-T5-Large** to test whether stronger text modeling alone improves compositional reward prediction. For this ablation, we use the *small* reward-head setting. We report changes in accuracy relative to the baseline fine-tuned on CMI-PREF.

As shown in Table 16, Flan-T5 provides a gain on one text-only Music Arena subset, is neutral on CMI text+lyrics, and degrades consistently when audio conditioning is present. This suggests that the current bottleneck is cross-modal audio-text fusion rather than text encoder capacity.

### F.3. Ablation Study on Mapping Functions

We investigate the impact of different mapping functions used to transform the predicted preference scores for MOS regression. We compare four settings:

- **None**: No transformation is applied.

- **Tanh**: The method adopted in our main experiments, using a Tanh activation to bound the output.

- **Linear**: A linear projection without Tanh activation.

- **Ordinal**: We employ an ordinal regression objective with learnable classification margins during training. Note that for evaluation, we use the model's raw scalar output to compute metrics, bypassing the learned margins.

To ensure a fair comparison, all models are initialized from the CMI-Pref-Pseudo pretraining checkpoint (2,000 steps) and finetuned on CMI-Pref and MusicEval. As shown in Table 17, we observe that the choice of mapping function does not yield significant performance differences.

Table 17. Aggregated results on mapping ablations

| Datasets Metrics ↑ | PAM Mean SRCC | MusicEval SRCC | Music Arena ACC | CMI-Pref Mean ACC |
|---|---|---|---|---|
| None | 0.5069 | 0.4659 | 73.73% | 75.3% |
| Tanh | 0.6024 | 0.4702 | 73.8% | 74.8% |
| Linear | 0.4595 | 0.4691 | 73.58% | 74.40% |
| Ordinal | 0.6079 | 0.4665 | 72.31% | 75.80% |

## F.4. The Effect of Duration on Model Inference

The variable-length nature of audio makes it important to analyze the effect of duration at inference time. Scoring an entire waveform can be memory-intensive, while restricting inference to short windows may reduce compute but risk missing long-range musical structure and text–music relationships. We evaluate three inference strategies using the same reward model (w/ f.t.: CMI + MusicEval):

1. *first 10*: Use only the first 10 seconds of audio to infer the Musicality and Alignment scores.

2. *mean 10*: Split the audio into non-overlapping 10-second chunks (hop size 10 seconds), infer scores for each chunk, and take the average as the final score.

3. *first 120*: Match the training setting by extracting MuQ embeddings from the first 120 seconds (concatenating four 30-second segments), and then infer scores using the learned weights.

Among these, *mean 10* is the most compute-intensive. With a maximum duration of 120 seconds and batched inference, its memory overhead is only slightly larger than *first 120*, but it requires more forward passes.

Table 18 reports accuracy on CMI-RewardBench subsets across duration bins. We do not report metrics on PAM and MusicEval due to their short duration($\leq 60$ for MusicEval and $\leq 30$ for PAM). On Music Arena, *first 120* achieves the best performance in most bins and shows clear gains over *first 10*, suggesting that short excerpts often under-represent

relevant musical content. *mean 10* attains comparable accuracy to *first 120* in several cases, indicating that aggregating local judgments can approximate longer-context inference. On CMI-Pref, the same trend generally holds but is less pronounced: *first 10* remains competitive, especially in shorter-duration bins, implying that some preference decisions can be resolved from early content.

Table 18. Accuracy (%) by audio duration across inference variants. Highest per bin is **bold**.

| Metric | Variant | Duration (s), Acc (%) | | | |
|---|---|---|---|---|---|
| | | [10,30) | [30,60) | [60,120) | [120,500] |
| Music Arena | *n* | *119* | *88* | *531* | *600* |
| | *first 10* | 72.3 | **53.4** | 64.2 | 59.5 |
| | *mean 10* | **75.6** | 48.9 | 75.7 | 67.0 |
| | *first 120* | **75.6** | **53.4** | **81.1** | **67.3** |
| CMI-Pref Musicality | *n* | *196* | *77* | *73* | *154* |
| | *first 10* | 77.6 | 71.4 | 87.7 | **76.0** |
| | *mean 10* | **78.1** | **76.6** | 89.0 | 74.7 |
| | *first 120* | 77.6 | **76.6** | **90.4** | 73.7 |
| CMI-Pref Alignment | *n* | *196* | *77* | *73* | *154* |
| | *first 10* | 68.4 | 74.0 | **83.6** | 66.9 |
| | *mean 10* | 70.4 | **75.3** | 80.8 | 69.5 |
| | *first 120* | **72.5** | **75.3** | **83.6** | **70.8** |

Tables 19 and 20 quantify how predicted scores change across inference strategies. We compare configurations using Linear Correlation Coefficient (LCC) and Root Mean Square Error (RMSE). We also report summary statistics of the reference scores using the standard *first 120* to contextualize absolute differences.

Across both Musicality and Alignment, increasing duration is associated with higher RMSE and lower LCC when comparing short-context to long-context predictions, indicating that short excerpts become less predictive of long-context scores as track length grows. When comparing *first 10* against *first 120*, the overall RMSE-to-STD ratios are 0.5575 (Musicality) and 0.6133 (Alignment), suggesting that duration mismatch introduces substantial variation relative to the natural score spread and motivating longer-context inference. Notably, Alignment exhibits lower short-vs-long correlation than Musicality, consistent with alignment requiring broader temporal context. Additionally, the correlation for short time and long time is lower for alignment, indicating that alignment needs longer context.

Finally, while *first 10* exhibits noticeably weaker agreement with *first 120* (lower LCC and higher RMSE), it still attains reasonably strong performance on several subsets, suggesting that short-context inference can be "good enough" for some downstream uses (e.g., providing a coarse preference signal during post-training). At the same time, the gap to *first 120* indicates that longer temporal context remains ben-

eficial, and we leave it to future work to develop reward models that more reliably leverage long-range structure and text–music relationships.

*Table 19.* Effect of inference duration on predicted Musicality (RMSE & LCC between three configurations). Statistics for *first 120* is shown

| Pair | [10,30) $n=1311$ | [30,60) $n=824$ | [60,120) $n=346$ | [120,500] $n=1639$ | Overall $n=4120$ |
|---|---|---|---|---|---|
| *RMSE* | | | | | |
| *f10* vs *f120* | 0.3022 | 0.3960 | 0.5465 | 0.7734 | 0.5687 |
| *m10* vs *f120* | 0.2289 | 0.2067 | 0.4292 | 0.5014 | 0.3751 |
| *f10* vs *m10* | 0.2836 | 0.3583 | 0.3485 | 0.4478 | 0.3758 |
| *LCC* | | | | | |
| *f10* vs *f120* | 0.9597 | 0.9318 | 0.8234 | 0.7541 | 0.8728 |
| *m10* vs *f120* | 0.9790 | 0.9837 | 0.9292 | 0.9271 | 0.9597 |
| *f10* vs *m10* | 0.9619 | 0.9445 | 0.8811 | 0.8153 | 0.9159 |
| *Ref (first120) score stats* | | | | | |
| mean±std | 0.92±1.06 | 1.11±1.09 | 1.71±0.75 | 1.87±0.74 | 1.40±1.02 |
| **[min, max]** | | | | | [-2.31, 4.50] |

*Table 20.* Effect of inference duration on predicted Text-Music Alignment (RMSE & LCC between three configurations). Statistics for *first 120* is shown

| Pair | [10,30) $n=1311$ | [30,60) $n=824$ | [60,120) $n=346$ | [120,500] $n=1639$ | Overall $n=4120$ |
|---|---|---|---|---|---|
| *RMSE* | | | | | |
| *f10* vs *f120* | 0.2726 | 0.3104 | 0.4164 | 0.6110 | 0.4539 |
| *m10* vs *f120* | 0.2084 | 0.1777 | 0.3082 | 0.3410 | 0.2727 |
| *f10* vs *m10* | 0.2568 | 0.2922 | 0.2928 | 0.4022 | 0.3311 |
| *LCC* | | | | | |
| *f10* vs *f120* | 0.9480 | 0.9255 | 0.8329 | 0.6836 | 0.8417 |
| *m10* vs *f120* | 0.9708 | 0.9764 | 0.9309 | 0.9062 | 0.9490 |
| *f10* vs *m10* | 0.9508 | 0.9367 | 0.8828 | 0.7738 | 0.8933 |
| *Ref (first120) score stats* | | | | | |
| mean±std | 1.05±0.85 | 1.27±0.80 | 1.54±0.62 | 1.54±0.55 | 1.33±0.74 |
| **[min, max]** | | | | | [-1.76, 3.72] |

# G. Discussions on Musicality

## G.1. Musicality on Music Arena

To understand how musicality and alignment jointly dictate real-world user preferences, we regress these two predicted dimensions against the overall human preference labels collected from Music Arena. Since Music Arena only provides a single, holistic preference label per pair, we evaluate whether users lean more towards aesthetic musicality or strict instructional alignment.

*Table 21.* Regression analysis of overall human preference on Music Arena using predicted Musicality (Mus) and Alignment (Ali) scores. The results indicate that general user preference is predominantly driven by musicality.

| Method | Accuracy (%) | AUC | MSE |
|---|---|---|---|
| **Single Metric Threshold (Threshold = 0)** | | | |
| Musicality Diff | **73.4** | 0.8001 | 0.1884 |
| Alignment Diff | 69.7 | 0.7714 | 0.2031 |
| Mus + Ali Diff (Equal) | 72.6 | 0.7962 | 0.1851 |
| **Regression Models (5-fold CV)** | | | |
| Logistic Regression | 73.1 | 0.7991 | 0.1845 |
| SVM-Linear | 72.5 | 0.7990 | 0.1846 |
| SVM-RBF | 72.6 | 0.7798 | 0.1875 |

Table 21 demonstrates that musicality has an overwhelmingly dominant influence on general preference. A simple thresholding of the musicality difference already yields strong performance, while regression models provide no gains (e.g., -0.3% in accuracy) due to fold-level noise.

This dominance is directly reflected in the fitted model weights:

- **Logistic Regression:** $\text{coef}_{\text{mus}} = 1.2296$, $\text{coef}_{\text{ali}} = 0.1991$, intercept $= 0.0592$.

- **Linear SVM:** $\text{coef}_{\text{mus}} = 1.2229$, $\text{coef}_{\text{ali}} = 0.1861$.

This near-optimal alignment with in-the-wild user behavior echoes our annotation protocols.

## G.2. Musicality: Intrinsic Audio Quality vs. Contextual Preference

In existing datasets and models such as SongEval (Yao et al., 2025), musicality is primarily treated as an absolute measure of "intrinsic audio quality". These metrics are typically *reference-free*, focusing exclusively on the audio itself. However, in the context of music generation, musicality often interacts—or even interferes—with the generation context, specifically the multimodal prompts fed into the model.

Compared to passive music listening, when generating music, a user's perception of whether a piece is "musical" is rarely isolated. Instead, it is inherently influenced by the aesthetic expectations established by their initial creative intent. Therefore, we argue that evaluating musicality in a practical AIGC workflow requires shifting from a context-free measurement to a *context-aware* judgment.

### G.2.1. EMPIRICAL OBSERVATIONS ON PROMPT CONTRIBUTION

To empirically validate whether prompt information assists in predicting human musicality preferences, we conducted an ablation study using the CMI-Pref dataset. We compared

two variants of our model: (1) a variant that drops all prompt conditions during both training and inference (effectively a reference-free evaluator), and (2) a variant utilizing regular conditional training.

As shown in Table 22, our experimental results reveal several key findings:

- **Performance Gain from Context:** Incorporating prompt conditions significantly improves the overall musicality prediction accuracy from 70.20% to 75.60% ($\Delta = +5.40\%$). Specifically, the text-only setting also shows a marginal gain (+1.60%).

- **Reference Audio as a Strong Anchor:** The improvement is most pronounced when reference audio is present. The "Text + Audio + Lyrics" and "Text + Audio w/o Lyrics" modalities yield substantial gains of +13.60% and +10.40%, respectively. This suggests that reference-based context provides a much clearer signal for aesthetic preference than text alone.

- **Modality Bottleneck in Lyrics:** Conversely, the "Text + Lyrics w/o Audio" setting is the only modality that shows a decline (-4.00%). We attribute this to the ineffectiveness of the frozen MuQ-MuLan text encoder, which struggles to process raw lyric structures without acoustic grounding, thereby introducing semantic noise rather than useful context.

- **Comparison with Reference-Free Baselines:** As shown in our main results, Table 2, reference-free metrics like SongEval-RM show competitive performance on general acoustic quality but are outperformed by our context-aware model on CMI-Pref. This highlights the limitation of condition-free evaluation in complex compositional tasks.

*Table 22.* Ablation study on prompt conditions for Musicality prediction on CMI-Pref. We compare models trained and evaluated with and without compositional conditions across different prompt modalities ($n = 125$ for each subset).

| Prompt Modality | w/ Condition Acc (%) | w/o Condition Acc (%) | $\Delta$ |
|---|---|---|---|
| Text + Audio + Lyrics | **82.40** | 68.80 | +13.60 |
| Text + Audio w/o Lyrics | **79.20** | 68.80 | +10.40 |
| Text + Lyrics w/o Audio | 72.00 | **76.00** | -4.00 |
| Text only | **68.80** | 67.20 | +1.60 |
| **Overall** ($N = 500$) | **75.60** | 70.20 | +5.40 |

### G.2.2. HYPOTHESES: WHY DOES CONTEXT ASSIST MUSICALITY PREDICTION?

We propose two primary hypotheses to explain why the inclusion of prompts—which are in theory independent of absolute audio quality—enhances the model's predictive performance:

**1. Inherent Association in Human Judgment:** During annotation, human experts may find it difficult to completely decouple musicality from alignment. For instance, a prompt specifying a "lo-fi aesthetic" might lead an annotator to perceive low-fidelity audio as a musical choice rather than a technical flaw. Thus, the prompt acts as a "taste anchor" that recalibrates the aesthetic scale.

**2. Inference Assistance via Non-trivial Shortcuts:** From the data correlation perspective, there is an undeniable, intrinsic correlation between the prompt and the evaluated audio, as the latter is directly generated conditioned on the former from a music generation model. Because of this inherent link, the prompt provides critical contextual clues about the expected acoustic features. By attending to the prompt, the model can anticipate the intended musical elements and more easily identify task-specific artifacts that degrade quality but are difficult to detect from the raw audio alone.

### G.2.3. IMPLICATIONS FOR EVALUATING MUSIC CREATION

Regardless of whether these gains stem from human psychological anchoring or model-side statistical correlation, the results highlight a crucial distinction between *music listening* and *music creation*. While standalone acoustic metrics are sufficient for evaluating isolated tracks, the evaluation of AIGC music creation fundamentally requires context. Therefore, **multimodal inputs (Text, Lyrics, Audio) should be treated as an integral part of the evaluator's state**. For reward models in these tasks, transitioning from absolute acoustic measurement to context-aware preference is essential for achieving closer alignment with real-world user intent.

## H. Statistical Details

### H.1. Significance of Benchmark Improvements

We report additional significance checks from the rebuttal for key preference-based gains. These tests compare our model with strong baselines on CMI-PREF and Music Arena. The following checks indicate that the main improvements are unlikely to be explained by chance:

1. **CMI-Pref musicality**: CMI-Pref vs Gemini2.5-Pro, 78.60% vs 70.00%, $p = 9.30 \times 10^{-4}$.

2. **CMI-Pref musicality**: CMI-Pref vs audiobox-PQ, 78.60% vs 73.80%, $p = 3.74 \times 10^{-2}$.

3. **Music Arena musicality**: CMI+MusicEval vs Gemini2.5-Pro, 73.21% vs 69.75%, $p = 2.37 \times 10^{-2}$.

4. **High-confidence musicality split**: CMI-Pref vs Gemini2.5-Pro, 85.71% vs 75.96%, $p = 1.09 \times 10^{-3}$.

*Table 23.* Significance checks for human listening results in test-time scaling.

| Backbone / Comparison | Ordering | $p$-value |
|---|---|---|
| Stable Audio | GT > Top-10 | 0.003 |
| MusicGen | GT > Top-1 | 0.0001 |
| Stable Audio | GT > Top-1 | 0.0001 |
| MusicGen | Top-10 > Top-3 | 0.016 |
| MusicGen | Top-3 > Top-1 | 0.101 |
| Stable Audio | Top-10 > Top-3 | 0.890 |
| Stable Audio | Top-3 > Top-1 | 0.239 |

## H.2. Human Listening Significance for Test-Time Scaling

For test-time scaling (Sec. 4.3), objective reranking uses 2,800 prompts from MusicGen-small and Stable-Audio-Open-small. Subjective listening uses **50 prompts** and **four ranks** (GT, Top-10, Top-3, Top-1), yielding 200 generations and 300 pairwise comparisons. Following the rebuttal protocol, ties are split equally and significance is computed with an exact one-sided binomial test on effective non-tied comparisons.

Table 23 shows that the largest ranking gaps are significant, while differences among reranked candidates are often not significant. This matches Fig. 3, where GT is consistently preferred and Top-3/Top-10 are much closer.

## I. Detailed Analysis of Results

### I.1. Lyrics Transcription as a Proxy Metric

We further test whether explicit lyric transcription can serve as a simple proxy for lyric-following preference. On the 250 lyric-conditioned pairs in CMI-Pref-test, we transcribe all 500 audio samples using Whisper ASR and compute word error rate (WER) against the prompt lyrics. A WER-based preference rule selects the candidate with lower WER; when both candidates obtain identical WER, the comparison is treated as a tie and excluded from WER-based decision accuracy, yielding 225 non-tie comparisons.

WER-based preference prediction reaches only 60.0% accuracy, substantially below the 76.0% accuracy of CMI-RM on the same lyric-conditioned setting. This result suggests that transcription quality is a useful auxiliary signal, but not a sufficient predictor of human preference for lyric-conditioned music. Human annotators also consider vocal naturalness, prosody, musicality, structural coherence, and how well the lyrics are integrated with the requested musical style, which are not captured by WER alone.

### I.2. Performance on Music Arena Subsets

Tables 24 and 25 reveal two dominant factors that influence reward model performance: temporal distribution shift and annotation confidence.

Table 24 shows a clear temporal drift in Music Arena. Several reference-free metrics, particularly the Audiobox variants, exhibit a noticeable decline in accuracy from early to later months, indicating sensitivity to evolving generation quality and user preference distributions. SongEval-RM demonstrates relatively stronger stability across time, while general-purpose multimodal LLMs show substantial month-to-month variance, suggesting limited robustness for fine-grained musicality judgment. In contrast, our models maintain more consistent performance over time and achieve improved robustness on vocal music, especially when jointly fine-tuned with MusicEval, highlighting the benefit of music-specific supervision.

### I.3. Performance on CMI-Pref Subsets

Table 25 analyzes performance on the CMI-Pref test set by stratifying samples according to annotator-reported confidence, which in our data collection protocol explicitly measures the perceived *preference margin* between two candidates. Unlike traditional pairwise annotation schemes that only record a binary choice, annotators are required to select a preferred sample and additionally indicate how strongly one sample is preferred over the other.

We observe a clear monotonic relationship between confidence and model accuracy across all methods. Higher-confidence comparisons, corresponding to larger perceptual gaps in musical quality or instruction adherence, are consistently easier to predict. Lower-confidence comparisons reflect fine-grained distinctions with smaller margins, which naturally impose a more challenging learning problem.

Importantly, our proposed reward models exhibit the largest performance gains in the high-confidence regime. The CMI-Pref fine-tuned model substantially outperforms all baselines when confidence is high, indicating strong alignment with clear and decisive human preferences. At low confidence levels, performance differences across methods narrow, suggesting that improvements on near-tie comparisons are fundamentally limited by the small preference margins rather than model capacity.

Overall, these results demonstrate that explicitly modeling preference strength during data collection provides a principled way to analyze reward model behavior beyond binary accuracy, and highlights the effectiveness of our approach in capturing dominant human preference signals relevant for downstream reranking and selection.

*Table 24.* Benchmark Musicality ACC Results on Music Arena (Time & Data Type Analysis).

| Model | Total (%) | Time Period (Month) | | | | | Data Type | |
| --- | --- | --- | --- | --- | --- | --- | --- | --- |
| | | Jul-Aug | Sep | Oct | Nov | Dec | Instru | Vocal |
| PAM score | 63.13 | 69.13 | 75.41 | 58.56 | 53.14 | 55.27 | 56.84 | 68.35 |
| audiobox-CE | 64.25 | 67.79 | 73.77 | 63.36 | 60.14 | 55.27 | 64.25 | 64.25 |
| audiobox-CU | 67.76 | 71.36 | 77.0 | 66.78 | 60.14 | 60.73 | 68.04 | 67.53 |
| audiobox-PC | 58.73 | 63.75 | 70.49 | 55.82 | 54.54 | 48.00 | 52.88 | 63.57 |
| audiobox-PQ | 67.54 | 68.01 | 74.86 | 67.47 | 65.04 | 63.27 | 68.04 | 67.12 |
| SongEval-RM | **73.88** | 73.60 | **78.69** | 71.92 | 64.34 | **78.18** | **77.27** | 71.08 |
| Omni-Reward | 54.02 | 49.44 | 49.18 | 60.61 | 53.85 | 57.81 | 54.53 | 53.62 |
| Qwen2-audio | 35.99 | 38.26 | 37.91 | 31.23 | 38.24 | 34.91 | 30.66 | 39.63 |
| Qwen2.5-Omni | 36.05 | 40.04 | 39.89 | 38.36 | 24.48 | 30.55 | 29.98 | 41.23 |
| Qwen3-Omni | 59.63 | 59.06 | 58.47 | 59.93 | 60.84 | 60.36 | 62.93 | 56.89 |
| Gemini2.5-Flash | 64.12 | 60.54 | 62.84 | 69.52 | 61.27 | 66.55 | 65.84 | 62.70 |
| Gemini2.5-Pro | 69.75 | 65.99 | 64.48 | 71.88 | 73.24 | 75.27 | 75.78 | 64.69 |
| Gemini3-Pro | 68.85 | 63.59 | 60.28 | 74.31 | 78.63 | 72.22 | 76.03 | 62.46 |
| - w/o f.t.: Distill only | 65.37 | 67.56 | 68.3 | 67.8 | 68.53 | 55.64 | 63.26 | 67.12 |
| - w/ f.t.: CMI-Pref | 71.57 | 71.36 | 75.41 | 72.95 | 65.04 | 71.27 | 73.81 | 69.71 |
| - w/ f.t.: CMI + MusicEval | 73.21 | **74.27** | 75.96 | **74.66** | 65.73 | 72.00 | 74.14 | **72.44** |

*Table 25.* Benchmark Musicality ACC Results on CMI-Pref (Ablation Analysis).

| Musicality | Total (500) | Confidence Level | | | Data Type | |
| --- | --- | --- | --- | --- | --- | --- |
| | | Conf < 3 (66) | Conf = 3 (128) | Conf > 3 (306) | Instru (250) | Vocal (250) |
| PAM score | 65.40% | 63.64% | 64.84% | 66.01% | 67.60% | 63.20% |
| audiobox-CE | 71.80% | 63.64% | 72.66% | 73.20% | 70.80% | 72.80% |
| audiobox-CU | 71.40% | 66.67% | 71.88% | 72.22% | 68.80% | 74.00% |
| audiobox-PC | 59.00% | 46.97% | 54.69% | 63.40% | 60.00% | 58.00% |
| audiobox-PQ | 73.80% | **74.24%** | **78.13%** | 71.90% | 73.20% | 74.40% |
| SongEval-RM | 72.40% | 63.64% | 71.09% | 74.84% | 70.80% | 74.00% |
| Omni-Reward | 65.60% | 56.06% | 66.41% | 67.32% | 59.20% | 72.00% |
| Qwen2-audio | 8.60% | 4.54% | 7.81% | 9.80% | 5.20% | 12.00% |
| Qwen2.5-omni | 17.40% | 15.15% | 16.41% | 18.40% | 15.20% | 19.60% |
| Qwen3-omni | 60.40% | 54.55% | 53.12% | 64.70% | 63.20% | 57.60% |
| Gemini2.5-flash | 64.20% | 56.57% | 57.94% | 71.94% | 74.40% | 54.00% |
| Gemini2.5-pro | 70.00% | 48.49% | 69.29% | 75.96% | **78.80%** | 61.20% |
| Gemini3-pro | 65.80% | 57.94% | 66.67% | 72.67% | 73.20% | 58.40% |
| - w/o f.t.: Distill only | 70.80% | 53.03% | 67.19% | 76.14% | 65.60% | 76.00% |
| - w/ f.t.: CMI-Pref | 77.80% | 66.67% | 76.56% | 80.72% | 74.00% | 81.60% |
| - w/ f.t.: CMI + MusicEval | 78.10% | 66.67% | 75.39% | 81.70% | 75.60% | 80.60% |

*Table 26.* Benchmark Text-Music Alignment ACC Results on CMI-Pref(Ablation Analysis).

| Text-Music Alignment | Total (250) | Confidence Level | | | Data Type | |
|---|---|---|---|---|---|---|
| | | Conf < 3 (47) | Conf = 3 (58) | Conf > 3 (145) | Instru (125) | Vocal (125) |
| audiobox-CE | 60.00% | 45.95% | 63.64% | 61.90% | 56.80% | 63.20% |
| audiobox-CU | 59.60% | 54.05% | 63.64% | 59.18% | 53.60% | 65.60% |
| audiobox-PC | 56.00% | 35.13% | 59.09% | 59.86% | 55.20% | 56.80% |
| audiobox-PQ | 59.60% | 54.05% | 65.15% | 58.50% | 55.20% | 64.00% |
| CLAP score | 62.40% | 51.35% | 59.09% | 66.67% | 60.80% | 64.00% |
| CLAP music score | 70.40% | 75.68% | 60.61% | 73.47% | 67.20% | 73.60% |
| MuQ-Mulan | 66.40% | 67.56% | 60.00% | 69.39% | 64.80% | 68.00% |
| CLAMP3 score | 62.80% | 64.86% | 60.61% | 63.27% | 63.20% | 62.40% |
| Omni-Reward | 68.80% | 57.14% | 65.96% | 72.67% | 67.20% | 70.40% |
| Qwen2-audio | 1.60% | 2.70% | 1.51% | 1.36% | 0.80% | 2.40% |
| Qwen2.5-Omni | 34.40% | 27.03% | 33.33% | 36.73% | 31.20% | 37.60% |
| Qwen3-Omni | 63.60% | 56.76% | 63.64% | 66.67% | 67.20% | 60.00% |
| Gemini2.5-Flash | 60.80% | 48.65% | 60.61% | 63.95% | 65.60% | 56.00% |
| Gemini2.5-Pro | 67.20% | 54.05% | 66.67% | 70.75% | 71.20% | 63.20% |
| Gemini3-Pro | 64.00% | 54.05% | 66.67% | 65.31% | 67.20% | 60.80% |
| - w/o f.t.: Distill only | 69.60% | 65.96% | 70.69% | 70.34% | 67.20% | 72.00% |
| - w/ f.t.: CMI-Pref | 68.80% | 55.32% | 74.14% | 71.03% | 64.80% | 72.80% |
| - w/ f.t.: CMI + MusicEval | 70.20% | 51.06% | 68.96% | 76.89% | 67.60% | 72.80% |

*Table 27.* Benchmark Audio-Music Alignment ACC Results on CMI-Pref (Ablation Analysis).

| Audio-Music Alignment | Total (250) | Confidence Level | | | Data Type | |
|---|---|---|---|---|---|---|
| | | Conf < 3 (42) | Conf = 3 (47) | Conf > 3 (161) | Instru (125) | Vocal (125) |
| OmniRewardModel | 68.80% | 57.14% | 65.96% | 72.67% | 67.20% | 70.40% |
| Qwen2-audio | 11.60% | 11.90% | 4.84% | 14.38% | 8.80% | 14.40% |
| Qwen2.5-Omni | 28.80% | 26.19% | 25.81% | 30.82% | 25.60% | 32.00% |
| Qwen3-Omni | 64.00% | 54.76% | 61.29% | 67.81% | 64.80% | 63.20% |
| Gemini2.5-Flash | 62.00% | 50.00% | 61.29% | 65.75% | 69.60% | 54.40% |
| Gemini2.5-Pro | 72.80% | 52.38% | 72.58% | 78.77% | 73.60% | 72.00% |
| Gemini3-Pro | 66.80% | 61.90% | 53.23% | 73.97% | 68.80% | 64.80% |
| - w/o f.t.: Distill only | 73.20% | 61.90% | 63.83% | 78.88% | 69.20% | 77.20% |
| - w/ f.t.: CMI-Pref | 79.20% | 66.67% | 72.34% | 84.47% | 76.00% | 82.40% |
| - w/ f.t.: CMI + MusicEval | 77.80% | 55.95% | 78.72% | 83.22% | 76.40% | 79.20% |

## J. Leaderboard for music generation models

We report an overall leaderboard of music generation models evaluated by our trained reward model, CMI-RM. All audio generations come from CMI-PREF-PSUEDO, where each model is queried with the same pool of prompts (and, when applicable, the same audio prompts). For every generated sample, we run CMI-RM with the prompt (text instruction and optional audio prompt) and the generated audio as inputs, and obtain two scalar signals: *alignment* and *musicality*.

To aggregate results into a unified ranking, we construct prompt-wise full round-robin comparisons: for each unique prompt (under a fixed input type), we enumerate all pairs of model outputs generated from that same prompt. For each input type (Inst./Song × w/ audio / w/o audio) and each dimension (alignment / musicality), CMI-RM assigns a binary label for each pair (i.e., whether model $a$ is preferred over model $b$ under the same prompt). We then compute

*Table 28.* Generation counts per model by modality

| Model | Instrumental | | Song | |
| --- | --- | --- | --- | --- |
| | w/ audio | w/o audio | w/ audio | w/o audio |
| *suno-v5* | 212 | 221 | 213 | 288 |
| *mureka-o2* | 0 | 0 | 15 | 163 |
| *suno-v4.5-plus* | 214 | 221 | 216 | 288 |
| *mureka-v7.5* | 7 | 117 | 8 | 211 |
| *minimax-music-2* | 0 | 0 | 27 | 429 |
| *suno-v4.5* | 165 | 193 | 163 | 260 |
| *suno-v3.5* | 0 | 399 | 0 | 378 |
| *suno-v4* | 193 | 193 | 184 | 260 |
| levo | 0 | 0 | 484 | 735 |
| magenta-rt-large | 2979 | 3037 | 0 | 0 |
| *satwo* | 445 | 320 | 0 | 0 |
| sao | 1348 | 4674 | 0 | 0 |
| *loudly-music* | 10 | 248 | 0 | 0 |
| acestep | 1346 | 4675 | 1470 | 1468 |
| audioldm | 829 | 937 | 0 | 0 |
| diffrhythm | 831 | 513 | 843 | 730 |
| musicldm | 1347 | 4678 | 0 | 0 |
| jamify | 580 | 341 | 496 | 455 |
| yue | 0 | 0 | 855 | 898 |
| sao-small | 830 | 938 | 0 | 0 |
| musicgen-medium | 830 | 937 | 0 | 0 |
| audioldm2-music | 1348 | 4677 | 0 | 0 |
| songgen | 0 | 0 | 2125 | 2130 |

a Bradley–Terry (BT) style ranking (equivalently, an Elo-like rating variant) by aggregating these pairwise win/loss labels, and report the resulting rank scores in Table 29. The corresponding ranking scores are reported in Table 30. Inst. denotes instrumental music generation, Song denotes song generation, and "w/ audio" indicates that an audio prompt is provided in addition to the text instruction; modalities not supported by a model are left blank. For readability, we linearly rescale scores as score $\times 400 + 1500$. Proprietary models (no public checkpoints) are marked in *italics*. Note that in Song (w/ audio), *mureka-o2* and *minimax-music-2* have fewer generations (see Table 28); hence their apparent advantage in this subset may be less reliable.

We observe that:

1. The gap between open-weight and closed-source models remains large: the top-5 models in each metric are dominated by proprietary systems.

2. In pure song generation, the gap between Suno and other proprietary models is narrowing. In modalities with abundant generations, Minimax Music and Mureka outperform several Suno variants.

3. Recent open-source models—LEVO for song generation, MAGENTA REALTIME for instrumental generation, and ACESTEP across both—exhibit competitive performance.

*Table 29.* Per-column ranks (1=best). Inst. refers to instrumental music generation, Song referes to song generation, w/audio indicates if an audio prompt is used. Unavailable modalities are left out.

| Model | Inst. (w/ audio) | | Inst. (w/o audio) | | Song (w/ audio) | | Song (w/o audio) | |
|---|---|---|---|---|---|---|---|---|
| | Alignment | Musicality | Alignment | Musicality | Alignment | Musicality | Alignment | Musicality |
| *suno-v5* | 1 | 1 | 3 | 3 | 3 | 3 | 2 | 3 |
| *mureka-o2* | - | - | - | - | 2 | 1 | 4 | 1 |
| *suno-v4.5-plus* | 2 | 2 | 1 | 2 | 4 | 4 | 3 | 5 |
| *mureka-v7.5* | 6 | 4 | 2 | 1 | 7 | 6 | 5 | 2 |
| *minimax-music-2.0* | - | - | - | - | 1 | 2 | 1 | 4 |
| *suno-v4.5* | 3 | 3 | 4 | 4 | 5 | 5 | 8 | 7 |
| *suno-v3.5* | - | - | 5 | 5 | - | - | 6 | 8 |
| *suno-v4* | 7 | 5 | 6 | 6 | 8 | 8 | 7 | 6 |
| levo | - | - | - | - | **6** | **7** | **9** | **9** |
| magenta-rt-large | **5** | **7** | **7** | **8** | - | - | - | - |
| *satwo* | 4 | 6 | 11 | 12 | - | - | - | - |
| sao | 8 | 9 | 8 | 10 | - | - | - | - |
| *loudly-music* | 12 | 10 | 9 | 7 | - | - | - | - |
| acestep | 11 | 11 | 13 | 11 | 9 | 9 | 10 | 11 |
| audioldm | 9 | 13 | 12 | 16 | - | - | - | - |
| diffrhythm | 13 | 12 | 10 | 9 | 10 | 10 | 12 | 10 |
| musicldm | 14 | 16 | 14 | 14 | - | - | - | - |
| jamify | 10 | 8 | 17 | 13 | 12 | 11 | 13 | 13 |
| yue | - | - | - | - | 11 | 12 | 11 | 12 |
| sao-small | 16 | 15 | 16 | 17 | - | - | - | - |
| musicgen-medium | 15 | 14 | 18 | 18 | - | - | - | - |
| audioldm2-music | 17 | 17 | 15 | 15 | - | - | - | - |
| songgen | - | - | - | - | 13 | 13 | 14 | 14 |

*Table 30.* Scores per column computed by Bradley-Terry ranking model. Scores are scaled by 400 and starts at 1500. Inst. refers to instrumental music generation, Song referes to song generation, w/audio indicates if an audio prompt is used. Unavailable modalities are left out.

| Model | Inst. (w/ audio) | | Inst. (w/o audio) | | Song (w/ audio) | | Song (w/o audio) | |
|---|---|---|---|---|---|---|---|---|
| | Alignment | Musicality | Alignment | Musicality | Alignment | Musicality | Alignment | Musicality |
| *suno-v5* | 1679.68 | 1722.06 | 1678.19 | 1736.51 | 1632.66 | 1629.18 | 1646.61 | 1640.35 |
| *mureka-o2* | - | - | - | - | 1635.26 | 1724.59 | 1617.19 | 1697.13 |
| *suno-v4.5-plus* | 1655.11 | 1695.81 | 1712.03 | 1749.49 | 1616.90 | 1613.08 | 1624.70 | 1625.19 |
| *mureka-v7.5* | 1557.91 | 1646.86 | 1711.13 | 1809.69 | 1585.47 | 1589.45 | 1602.01 | 1672.76 |
| *minimax-music-2.0* | - | - | - | - | 1640.14 | 1632.54 | 1665.48 | 1634.73 |
| *suno-v4.5* | 1623.63 | 1672.01 | 1635.68 | 1682.98 | 1610.11 | 1610.45 | 1578.10 | 1601.62 |
| *suno-v3.5* | - | - | 1600.23 | 1682.56 | - | - | 1583.51 | 1595.66 |
| *suno-v4* | 1542.72 | 1593.45 | 1597.78 | 1681.80 | 1526.98 | 1514.34 | 1580.83 | 1617.20 |
| levo | - | - | - | - | **1590.79** | **1576.12** | 1518.99 | 1504.16 |
| magenta-rt-large | 1577.06 | 1547.47 | 1535.96 | 1472.59 | - | - | - | - |
| *satwo* | 1597.36 | 1572.05 | 1438.93 | 1405.22 | - | - | - | - |
| sao | 1491.48 | 1475.49 | 1485.93 | 1439.23 | - | - | - | - |
| *loudly-music* | 1444.07 | 1472.19 | 1461.57 | 1475.87 | - | - | - | - |
| acestep | 1445.90 | 1446.98 | 1426.37 | 1427.27 | 1453.36 | 1429.65 | 1430.32 | 1383.79 |
| audioldm | 1480.04 | 1420.77 | 1429.10 | 1311.51 | - | - | - | - |
| diffrhythm | 1442.14 | 1428.54 | 1457.82 | 1456.67 | 1363.93 | 1364.21 | 1379.93 | 1386.05 |
| musicldm | 1389.73 | 1331.83 | 1404.59 | 1356.01 | - | - | - | - |
| jamify | 1471.97 | 1482.84 | 1350.61 | 1395.26 | 1351.20 | 1355.35 | 1216.11 | 1235.35 |
| yue | - | - | - | - | 1354.09 | 1324.13 | 1392.80 | 1334.58 |
| sao-small | 1385.25 | 1361.97 | 1351.36 | 1305.54 | - | - | - | - |
| musicgen-medium | 1388.95 | 1362.58 | 1329.69 | 1271.37 | - | - | - | - |
| audioldm2-music | 1326.99 | 1267.09 | 1393.03 | 1340.44 | - | - | - | - |
| songgen | - | - | - | - | 1139.10 | 1136.92 | 1163.41 | 1071.42 |

# K. LLM-as-a-Judge Prompt Architecture

To leverage frontier multimodal Large Language Models (MLLMs) as evaluators for generated music, we designed a structured message pipeline that seamlessly integrates text descriptions, optional lyrics, optional reference audio, and the generated audio candidates. This ensures the MLLM assesses the candidates based on comprehensive compositional multimodal instructions (CMI) and outputs a rigorously structured JSON response.

## K.1. Message Pipeline Construction

The input to the MLLM API is constructed sequentially as a list of messages. This pipeline dynamically adapts to the presence of optional modalities (lyrics, reference audio) and experimental settings (few-shot prompting, chain-of-thought reasoning). The exact message sequence is defined as follows:

1. **Task Instruction (System)**
   Initializes the evaluator role, defines the evaluation criteria (musicality, instruction following), and specifies the expected JSON schema.
   - **Role:** system
   - **Content:** [See Section K.2 for specific System Prompts]

2. **Few-Shot Examples (System, Optional)**
   If few-shot evaluation is enabled, an instruction and a set of predefined examples are appended to the message sequence. However, we disable this option during distillation, since they did not yield better consistency with human preference labels in our tests and led to a substantial increase in context length.
   - **Role:** system
   - **Content:** "You will now be shown several examples of audio comparisons to help you understand how to evaluate the audios." *(Followed by few-shot message pairs)*

3. **User Instruction Context (User)**
   Injects the specific CMI used for generation.
   - **Role:** user
   - **Content:**
     – [Text] "The user prompt was:<Prompt Text>"
     – [Text] *(If lyrics provided)* "The user provided the following lyrics: <Lyrics Text>"
     – [Audio] *(If audio provided)* <Base64 Encoded Reference Audio>

4. **Candidate Audios (User)**
   Presents the generated audio samples from the models being evaluated.

   - **Role:** user
   - **Content:**
     – [Text] "Now, listen to first music file that model 1 generated:"
     – [Audio] <Base64 Encoded Model A Audio>
   - **Role:** user
   - **Content:**
     – [Text] "Now, listen to second music file that model 2 generated:"
     – [Audio] <Base64 Encoded Model B Audio>

5. **Reasoning Instruction (System, Optional)**
   If chain-of-thought rationales are required for the dataset (e.g., CMI-Pref generation), an explicit reasoning instruction is appended.

   - **Role:** system
   - **Content:** "Explain your choice, no less than 200 words"

6. **Output Formatting Reminder (System, Optional)**
   For few-shot settings, a strict reminder of the JSON structure is appended to prevent output drift.

   - **Role:** system
   - **Content:** "Now, return your result. Remember, your evaluation format should be in json, [JSON Schema Reminder]"

**Implementation Note:** All API requests enforce the output format by setting response_format={"type": "json_object"} to guarantee parsable results.

## K.2. System Prompts

We utilized distinct system prompts depending on the granularity of the evaluation task.

PROMPT 1: OVERALL PREFERENCE EVALUATION

Used for determining the holistic quality of a generated sample, combining both audio quality and prompt adherence into a single assessment.

> You are an experienced critic familiar with music composition. You will have a text input, an optional audio input, both consists as the user prompt. You will then be sent 2 music audios generated by two models. Your task is to select which model's music is better.
>
> Your answer should be in 'model_a', 'model_b' or 'both', 'neither' for your choice.
>
> - model_a means the first audio is better.
> - model_b means the second audio is better.
> - both means they are equally good and hard to distinguish.
> - neither means both are bad.

Besides this choice, you should also give a score to access the overall quality for each audio on a scale of 1-10, where 1 means terrible and 10 means excellent.

- Please note that current ai-generated music usually has some limitations, so it's rare to a high score like 9 or 10.
- Besides your general impression, scoring should also take basic musical elements into consideration, such as:
  - recording quality (no distortion and scratches)
  - instrument validity (does it sound like the real instrument, where ai commonly struggles with orchestral instruments and traditional instruments)
  - genre consistency (does the music fit the requested genre, where there might a request for a blend)
  - rhythm (stability and creativity)
  - melody (creativity and also stability, is there a clear musical idea)
  - musical structure (repetition and variation, does the music evolve over time if it's a long piece)
- Each failure in one of these aspects should deduct points from the total score.

"overall_preference": "model_a/model_b/both/neither",
"score_a": from 1-10,
"score_b": from 1-10

PROMPT 2: MULTI-DIMENSIONAL EVALUATION (MUSICALITY & INSTRUCTION FOLLOWING)

Used to disentangle the evaluation into two distinct dimensions: musical quality and prompt alignment.

You are an experienced critic familiar with music composition. You will have a text input, an optional audio input, both consists as the user prompt. You will then be sent 2 music audios generated by two models.

Your task is to select which model's music is better from two aspects: which model follows the prompt better (Instruction Following) and which model produces more enjoyable music (Music Quality). For each aspect, your answer should be a choice and two scores.

For both instruction following and music quality, your answer should be in 'model_a', 'model_b' or 'both', 'neither' for your choice.

- model_a means the first audio is better.
- model_b means the second audio is better.
- both means they are equally good and hard to distinguish.
- neither means both are bad.

Besides this choice, you should also give a score on each aspect for each audio on a scale of 1-10, where 1 means terrible and 10 means excellent. You can give float scores like 7.5, the max precision is 1 decimal place.

- Please note that current ai-generated music usually has some limitations, so it's rare to a high score like 9 or 10.
- Besides your general impression, scoring should also take basic musical elements into consideration, such as genre, rhythm, melody, arrangement, timbre, and musical structure.

For **music quality**, you can consider:

- recording quality (no distortion and scratches)
- instrument validity (does it sound like the real instrument, where ai commonly struggles with orchestral instruments and traditional instruments)
- rhythm (stability and creativity)
- melody (stability and creativity)
- arrangement (stability: are the instruments appearing in a logical way, creativity: are there interesting musical ideas)
- musical structure (repetition and variation, does the music evolve over time if it's a long piece)
- Each failure in one of these aspects should deduct points from the score.

For **instruction following**, you can consider:

- genre consistency (does the music fit the requested genre, where there might a request for a blend)
- rhythm consistency (does the rhythm match the prompt, e.g., fast/slow, while maintaining musicality)
- melody consistency (does the melody style match the prompt, e.g., joyful/sad, while maintaining musicality)
- arrangement consistency (does the arrangement style match the prompt, e.g., correctly using requested instruments no more or less)
- Each failure in one of these aspects should deduct points from the score.

Your final answer should be in json format, writing all keys.

"music_quality": "model_a/model_b/both/neither",
"instruction_following":
"model_a/model_b/both/neither",
"MQ_score_a": from 1-10, one decimal place allowed, music quality score for model a,
"MQ_score_b": from 1-10, one decimal place allowed, music quality score for model b,
"IF_score_a": from 1-10, one decimal place allowed, instruction following score for model a,
"IF_score_b": from 1-10, one decimal place allowed, instruction following score for model b

PROMPT 3: RATIONALE GENERATION VARIANT

When generating pseudo-labels for distillation and interpretability is required, we replace the final JSON instruction in Prompt 2 with the following to explicitly elicit chain-of-thought rationales:

Finally, provide a detailed explanation of your evaluation, covering the strengths and weaknesses of each audio in relation to the prompt and the characteristics of the models. Your final answer should be in json format, writing all keys.

"music_quality": "model_a/model_b/both/neither",
"instruction_following":
"model_a/model_b/both/neither",
"MQ_score_a": from 1-10, one decimal place allowed, music quality score for model a,
"MQ_score_b": from 1-10, one decimal place allowed, music quality score for model b,
"IF_score_a": from 1-10, one decimal place allowed, instruction following score for model a,
"IF_score_b": from 1-10, one decimal place allowed, instruction following score for model b,
"reason": "your detailed explanation of the choice, less than 200 words"

## PROMPT 4: MUSICALITY EVALUATION (STRICT JSON VARIANT)

To ensure strict compliance with the required JSON output format and discrete 1–5 scoring scale, we adopt a simplified prompt for general-purpose AudioLLMs, which may otherwise struggle with complex instruction following.

> You are a music comparison judge. Compare Audio A and Audio B, then output ONLY scores.
>
> YOUR ONLY TASK: Give numerical scores (1-5) for each audio and pick a winner.
>
> OUTPUT FORMAT - You MUST output EXACTLY this JSON structure:
> {"overall_preference": "model_a", "score_a": 3, "score_b": 2, "reason": "brief reason"}
>
> MANDATORY RULES:
>
> - score_a MUST be a number from 1 to 5 (1=worst, 5=best)
> - score_b MUST be a number from 1 to 5 (1=worst, 5=best)
> - overall_preference MUST be one of: "model_a", "model_b", "both", "neither"
> - You MUST give scores. Scores of 0 are FORBIDDEN.
> - DO NOT output music_genre, music_description, key, tempo, or any music metadata.
> - DO NOT describe what the music sounds like.
> - ONLY output the JSON with scores.
>
> EXAMPLES:
> {"overall_preference": "model_a", "score_a": 4, "score_b": 2, "reason": "A has better quality"}
> {"overall_preference": "model_b", "score_a": 2, "score_b": 5, "reason": "B sounds cleaner"}
> {"overall_preference": "both", "score_a": 3, "score_b": 3, "reason": "similar quality"}
>
> Now compare the two audios and output your scores in JSON format:

## PROMPT 5: TEXT-MUSIC ALIGNMENT EVALUATION (STRICT JSON VARIANT)

Similarly, for evaluating how well the generated audios match the textual prompts under strict formatting constraints, we use:

> You are a music comparison judge. Compare how well Audio A and Audio B match the given text prompt.
>
> YOUR ONLY TASK: Give numerical scores (1-5) for how well each audio matches the prompt, then pick a winner.
>
> OUTPUT FORMAT - You MUST output EXACTLY this JSON structure:
> {"overall_preference": "model_a", "score_a": 3, "score_b": 2, "reason": "brief reason"}
>
> MANDATORY RULES:
>
> - score_a MUST be a number from 1 to 5 (1=worst match, 5=best match)
> - score_b MUST be a number from 1 to 5 (1=worst match, 5=best match)
> - overall_preference MUST be one of: "model_a", "model_b", "both", "neither"
> - You MUST give scores. Scores of 0 are FORBIDDEN.
> - DO NOT output music_genre, music_description, key, tempo, or any music metadata.
> - DO NOT describe what the music sounds like.
> - ONLY output the JSON with scores.
>
> EXAMPLES:
> {"overall_preference": "model_a", "score_a": 4, "score_b": 2, "reason": "A matches prompt better"}
> {"overall_preference": "model_b", "score_a": 2, "score_b": 5, "reason": "B follows the style"}
>
> Now compare the two audios and output your scores in JSON format:

## PROMPT 6: AUDIO-MUSIC ALIGNMENT EVALUATION (STRICT JSON VARIANT)

For evaluating style transfer and adherence to reference audio conditions, we adapt the strict prompt as follows:

> You are a music comparison judge. Compare how well Audio A and Audio B match the reference audio style.
>
> YOUR ONLY TASK: Give numerical scores (1-5) for how well each audio matches the reference, then pick a winner.
>
> OUTPUT FORMAT - You MUST output EXACTLY this JSON structure:
> {"overall_preference": "model_a", "score_a": 3, "score_b": 2, "reason": "brief reason"}
>
> MANDATORY RULES:
>
> - score_a MUST be a number from 1 to 5 (1=worst match, 5=best match)

- score_b MUST be a number from 1 to 5 (1=worst match, 5=best match)
- overall_preference MUST be one of: "model_a", "model_b", "both", "neither"
- You MUST give scores. Scores of 0 are FORBIDDEN.
- DO NOT output music_genre, music_description, key, tempo, or any music metadata.
- DO NOT describe what the music sounds like.
- ONLY output the JSON with scores.

EXAMPLES:
{"overall_preference": "model_a", "score_a": 5, "score_b": 3, "reason": "A matches reference better"}
{"overall_preference": "model_b", "score_a": 2, "score_b": 4, "reason": "B captures the style"}

Now compare the two audios and output your scores in JSON format:

