# OpenReview forum: "CMI-RewardBench: Evaluating Music Reward Models with Compositional Multimodal Instruction"
_ICML.cc/2026/Conference — ICML 2026 regular_

### Official Review · Reviewer_abPN · 2026-03-10

**Soundness:** 3
**Presentation:** 3
**Significance:** 3
**Originality:** 2
**Overall Recommendation:** 4
**Confidence:** 3

**Summary:**

This paper argues that the current evaluation mechanisms for music generation are insufficient. Recent music generation models use multimodal input, while evaluation methods focus on only one modality or a single aspect of quality. They first collect a large preference pseudo-labeled dataset (110k) and a smaller human-annotated preference dataset (4k). After that, they trained a Compositional Music Instruction reward model (CMI-RM) that considers combinations of text, lyrics, and reference audio, which predicts scores for musicality (MUS) and alignment (ALI). They prove that the proposed CMI-RM surpasses the baselines.

**Compliance With Llm Reviewing Policy:**

Affirmed.

**Final Justification:**

The authors' rebuttal fully answered my questions.

**Key Questions For Authors:**

1. Will the dataset be publicly released in the future? Or maybe just show a few samples and the model's prediction.
2. The paper argues that the reward model should agree with human preference, but how do you think the reward model will also agree with other metrics (e.g., beat alignment, emotion alignment)
3. If a music generation model uses your CMI-RM to optimize, do you think your model is robust enough? Or how to prevent reward hacking?

**Limitations:**

yes

**Strengths And Weaknesses:**

# Strengths
1. Clear motivation, the evaluation gap indeed needed to be addressed.
2. The 4k human-annotated preference dataset is a contribution to the music AI community (if open-sourced).
3. The proposed reward model surpasses other baselines.
4. The experiments are comprehensive, including multiple baseline models

# Weakness
1. No music samples demonstrated.
2. The proposed CMI-RM architecture is relatively simple, largely relying on frozen multimodal encoders and a scoring head.
3. Limited discussion about the interaction of different modalities. For Table 3, it's good to see how the proposed reward model performs in different subsets of CMI-Pref, but I think it will also be interesting if we can see how the reward model performs in the same subset of CMI-Pref, but with different combinations of text, lyrics, and audio (i.e, choose the subset with all conditions available, and see prediction results with different combination of modalities.)

---

> ### Author Rebuttal · Authors · 2026-03-31
>
> We thank the reviewer for the constructive feedback. We are glad that the reviewer recognizes the motivation, the value of the human-annotated dataset, and the strong empirical performance of CMI-RM. We address the main concerns below.
>
>
> ### 1. Samples and Release plan:
> All our data (CMI-Pref-Psuedo, CMI-Pref) are publically released under CC-BY-NC-SA-4.0 lisence. We also release the code for reward model inference and CMI-RewarBench Evaluation.
>
> Here is a link to preview some samples and the prediction results of our model on CMI-Pref-Test: https://anonymous.4open.science/w/cmi_demo/
>
> ### 2. On the “simple” architecture.
> We agree that the architecture is lightweight by design. Our goal is not architectural complexity, but a unified and parameter-efficient evaluator that supports heterogeneous optional inputs (text, lyrics, and reference audio) within a single model, rather than separate evaluators for different settings.
>
> Despite its simplicity, this design is effective: on the CMI-Pref test set, the CMI-Pref-finetuned model reaches 78.6% musicality accuracy, compared with 72.4% for SongEval-RM and 70.0% for Gemini 2.5 Pro.
>
> ### 3. Modality interaction / ablation.
>
> We thank the reviewer for this suggestion. To study modality interaction, we perform an inference-time ablation on the same 125 samples where all three conditions are available, by selectively dropping text, lyrics, and/or reference audio.
> The binary value indicates whether a modality exist.
>
> | Row order(Text,Lyrics, Audio) | alignment_acc | musicality_acc |
> | - | - | - |
> |                         0,0,0 |        0.760 |         0.804 |
> |                         0,0,1 |        0.784 |         0.824 |
> |                         0,1,0 |        0.784 |     **0.832** |
> |                         0,1,1 |        0.784 |         0.820 |
> |                         1,0,0 |        0.764 |         0.812 |
> |                         1,0,1 |      *0.792* |         0.808 |
> |                         1,1,0 |    **0.796** |       *0.824* |
> |                         1,1,1 |      *0.792* |         0.816 |
>
> For alignment, the main takeaway is that the model performs well across multiple input conditions. Multi-condition inputs consistently achieve strong results, with about 3% gain over the weakest setting, while the gap between the best two-modality setting and the full three-modality setting is below 1%. This suggests that compositional conditioning is helpful overall.
>
> For musicality, the inference-time differences are smaller, which is natural because musicality depends more directly on the generated audio itself. However, prompt modalities still help during training: compared with an audio-only reward model trained with the same achitecture, training with compositional conditions improves overall musicality prediction on CMI-Pref by +5.4% on 500 samples, only degrading on Text-Lyrics alignment for -4%.
>
> ### 4. Relation to other metrics
>
> We view human-preference prediction and fine-grained controllability metrics as complementary rather than interchangeable. Our benchmark focuses on agreement with human judgments over musicality and multimodal instruction following.
>
> We also tested WER as a proxy for lyric alignment.  WER-based prediction achieves only 60% accuracy on lyrics samples, CMI-RM, suggesting that fine-grained automatic alignment metrics do not yet fully capture human preference in this setting.
>
> We agree that beat alignment, emotion alignment, and symbolic-rule-based evaluation are valuable future directions, and we will clarify this distinction in the revision.
>
> ### 5. Robustness and reward hacking.
>
> We agree that reward hacking is important. Our paper focuses on building and validating a reward model and benchmark, rather than a full RL pipeline, so we do not claim that CMI-RM alone solves this issue. Promising directions include (1) cross-metric or hold-out evaluation during optimization; (2) hybrid rewards that combine CMI-RM with auxiliary music metrics such as aesthetic predictors or CLAP-based scores, as also explored in recent visual RL work (e.g., Dance-GRPO); and (3) explicitly studying overfitting to a single reward, since recent RL studies show that even strong reward models can be exploited without additional safeguards (e.g., Flow-GRPO; PrismAudio). We therefore view robust reward optimization for music generation as an important future direction, but orthogonal to the benchmark and evaluator contribution of this paper.
>
> Overall, we believe these additions strengthen the paper by clarifying that the main contribution is not merely a lightweight model, but a **unified reward modeling framework, benchmark, and dataset** for compositional multimodal music evaluation.

---

> > ### Author Rebuttal · Reviewer_abPN · 2026-04-03
> >
> > The rebuttal is convincing, I will raise my score

---

> > > ### Author Response · Authors · 2026-04-04
> > >
> > > We sincerely thank the reviewer for the positive follow-up and supportive assessment. We appreciate the recognition of the paper’s motivation and ecosystem value, and we will ensure that the revised manuscript incorporates the clarified release plan, modality-ablation discussion, and robustness framing more explicitly.

---

### Official Review · Reviewer_VKFf · 2026-03-12

**Soundness:** 3
**Presentation:** 3
**Significance:** 2
**Originality:** 3
**Overall Recommendation:** 4
**Confidence:** 5

**Summary:**

This paper introduces a framework for training and evaluating reward models for music generation under compositional multimodal instructions involving text, lyrics, and reference audio. Its main contributions are a large pseudo-labeled preference dataset (CMI-Pref-Pseudo, 110k pairs), a smaller human-annotated preference dataset (CMI-Pref, 4,027 pairs), a unified benchmark (CMI-RewardBench) that combines PAM, MusicEval, Music Arena, and CMI-Pref, and a compact parameter-efficient reward model family (CMI-RM, around 30M parameters) designed to operate across heterogeneous input settings. Experiments show that the proposed models achieve strong correlation with human judgments and provide practical gains for best-of-N reranking, while matching or outperforming specialized baselines and general-purpose multimodal language models on both regression- and preference-based evaluations.

**Compliance With Llm Reviewing Policy:**

Affirmed.

**Key Questions For Authors:**

The paper would benefit from a more detailed description of the human A/B evaluation used for test-time scaling. In particular, it would be useful to report the number of prompts, the number of candidate clips per prompt, the number of annotators per comparison, the treatment of ties, the randomization protocol, and any quality-control procedures. Confidence intervals and inter-rater agreement statistics would also make the conclusions easier to interpret.

A related question concerns the possible transfer of bias from the pseudo-labeling model. Since a large portion of the training data is derived from Qwen3-Omni, did the authors explicitly assess how much the final reward model depends on these pseudo labels, for example by comparing training on human-only, pseudo-only, and mixed subsets beyond the current ablations? It would also be interesting to know whether using an ensemble of annotator models was considered as a way to reduce judge-specific bias.

The training schedule also deserves further explanation. Given the dataset sizes, the reported number of pretraining and fine-tuning steps appears relatively small, so it would be helpful to clarify the rationale behind this choice and whether longer training improves or degrades performance, for example through overfitting to pseudo labels or benchmark-specific patterns.

The sensitivity of the approach to the choice of encoders is another important point. Since the architecture freezes MuQ-MuLan encoders, did the authors compare against alternatives such as CLAP-style audio encoders or text encoders that are more specialized for lyrics, and if so, how did those choices affect alignment quality?

For lyric-conditioned evaluation, it would also be useful to know whether the authors considered more explicit lyric-following baselines, such as ASR- or G2P-based alignment signals, to complement the human-judgment-based evaluation.

To improve reproducibility, the paper should also provide the prompt templates and instructions used for the multimodal judge baselines. Relatedly, did the authors explore chain-of-thought-style prompting or justification-based prompting to improve zero-shot judging performance?

The paper briefly touches on audio length, but more analysis would be valuable. How do the results vary across different durations, and across vocal versus instrumental subsets? Are there specific failure modes, such as dense polyphony, unusual meter, or weak lyric intelligibility, where the reward model becomes less reliable?

Given the known risks of underspecification and reward hacking in reward modeling, it would also be helpful to know whether the authors evaluated ensembles of reward models, for example across different random seeds or encoder choices, as a robustness measure.

For the Music Arena setting, more detail is needed on how the method handles temporal drift and prompt-distribution mismatch between training and testing. If any domain adaptation or calibration experiments were conducted, those would be useful to report.

Finally, since some evaluation components appear to rely on restricted or commercial generations, the paper would benefit from a clearer release plan. In particular, do the authors intend to provide an evaluation server or private test set to enable fair comparison by the broader community?

**Limitations:**

The paper discusses some limitations implicitly, but the treatment could be strengthened further. In particular, the authors should more explicitly acknowledge the dependence on pseudo-labels produced by a single proprietary multimodal model, the possibility of bias transfer from that judge into the reward model, the limited statistical reporting for some key human evaluations, and the partial reproducibility constraints introduced by TOS-restricted components. It would also be useful to discuss how these limitations may affect the generality and reliability of the benchmark and of the learned reward models.

Regarding societal impact, the paper would benefit from a more direct discussion of both benefits and risks. On the positive side, a unified benchmark and compact reward models could improve evaluation quality and reduce reliance on opaque, general-purpose judge models. At the same time, better reward models for music generation could enable large-scale optimization of generated content toward human preferences in ways that amplify stylistic imitation, reinforce annotator or model biases, or facilitate mass production of derivative music. The authors should also comment on copyright, authorship, and fair-use concerns, especially when lyrics and reference audio are used as conditioning signals. A concise limitations and societal impact section covering these points would strengthen the paper.

**Strengths And Weaknesses:**

The paper has several notable strengths. Its main contribution is to frame music reward modeling in a more realistic compositional multimodal setting, where evaluation is not limited to text prompts but can also incorporate lyrics and reference audio. This is a meaningful extension of the problem and reflects how music generation systems are increasingly conditioned in practice. Another strong aspect is the design of a unified and parameter-efficient reward model that can handle heterogeneous input configurations within a single architecture, rather than relying on separate task-specific models. The use of learnable null embeddings to accommodate optional modalities is elegant and makes the framework practically flexible. The pseudo-labeling pipeline is also thoughtfully designed, particularly the position-consistency filtering intended to reduce positional bias from the annotating multimodal model.

The experimental section is generally strong. The paper evaluates both regression-style and pairwise-preference settings across multiple datasets, reports several relevant rank- and correlation-based metrics, and includes informative subset analyses and training-mixture ablations. These experiments help clarify the contribution of pseudo-labeled versus human-labeled data and suggest that the two sources provide complementary signals. The best-of-N reranking experiments are also practically valuable, as they demonstrate that the learned reward models are not only predictive in offline evaluation but also useful at inference time. The paper is also fairly clear in its task formulation and architecture description, and the dataset statistics and inter-annotator agreement reporting are a positive aspect. Overall, the work addresses an important gap in music generation evaluation, and the relatively small reward model footprint increases its potential practical impact.

At the same time, the paper has some limitations. A central concern is the heavy reliance on pseudo-labels generated by a single proprietary multimodal model. Although the subsequent fine-tuning on human data helps mitigate this to some extent, there remains a risk that the reward model inherits systematic biases or blind spots from the pseudo-labeler. This concern is particularly relevant because the same family of judge models also appears in the evaluation context, and the paper does not fully analyze the possibility of contamination or unfair advantage. Another technical limitation is that the reward model freezes pretrained encoders, which is computationally efficient but may restrict the model’s ability to capture more subtle aspects of lyric alignment, long-range musical structure, or timbral and stylistic nuance. The reported training schedules also appear relatively short, and the paper does not provide much justification regarding convergence or stability.

There are also some experimental gaps. While the baseline coverage is reasonably broad, it could be stronger with additional recent open audio-text evaluation models and more explicit lyric-following baselines. The paper would also benefit from more complete reporting of statistical significance and confidence intervals, especially for the preference-based evaluations. In addition, the human evaluation details for inference-time scaling are somewhat sparse, which makes it harder to assess the reliability of those findings. Some presentation issues also remain, including minor editorial artifacts and occasional ambiguity in dataset descriptions and evaluation setup. Finally, the paper could better situate itself relative to recent work on multi-reward optimization and reward misspecification in music generation, especially since these issues are directly relevant to the proposed framework and its potential downstream use in RLHF-style loops.

---

> ### Author Rebuttal · Authors · 2026-03-31
>
> We thank the reviewer for the constructive feedback. We agree the key issues are pseudo-label bias, training stability, statistical reporting, and reproducibility. In the final version, we will clarify data/evaluation setup, add prompt templates and confidence intervals, and expand limitations and societal impact.
>
> ### 1. Dependence on Qwen3-Omni pseudo labels
> Our RM is not simply copying the pseudo-labeler. On 500 CMI-Pref-test samples, Qwen3-Omni is highly self-consistent (94.4% musicality / 94.1% alignment), but its agreement with our pseudo-pretrained model is only 80.6 / 79.2%, and drops to 63.7% / 68.4% after human finetuning. This shows human supervision shifts the decision boundary away from the teacher. Distill+CMI-Pref and Distill+Both also outperform Qwen3-Omni on MusicArena, improving accuracy from <60% to >70% with 1,340 human labels. Adding a second pseudo source (40k Gemini samples) further changes the model (agreement with pretrained drops to 65.2%/67.3%) but yields only minor final gains.
>
> ### 2. Training schedule and why it is short
> The short schedule is deliberate. Pretraining on CMI-Pref-Pseudo converges around 10k steps while most of the optimization gain is captured within the first 2000 steps. Longer pretraining keeps reducing pseudo-label validation loss, but cross-entropy on held-out human labels can rise above 1.2, far worse than the random baseline of $-\ln 0.5 = 0.693$. This indicates growing overconfidence under distribution shift rather than simple in-domain overfitting. Label smoothing largely fixes this overconfidence inside pseudo pretraining and makes checkpoint choice less sensitive. Finetuning shows the usual small-data overfitting regime: validation loss rises around 1k steps, so we early-stop near 250 steps. In short, pretraining is limited by calibration/transfer, whereas finetuning is limited by classic overfitting.
>
> ### 3. Test-time scaling protocol and statistical reporting
> We will expand the protocol in the final version. Objective reranking is computed on 2,800 prompts with generations from MusicGen and StableAudio. For subjective evaluation, we use 50 prompts and compare 4 ranking positions (GT, Top10, Top3, Top1), yielding 200 generations and 300 pairwise comparisons; ties are scored as 0.5 per side. We currently do not have inter-annotator agreement, so we present this as supporting evidence. GT > Top10 is significant (p<0.001), and same in MusicGen Top10 > Top3 (p=0.01) .
>
> In the final paper, we will add 95% bootstrap confidence intervals to all main accuracy metrics and reserve formal significance tests for primary comparisons against the strongest baseline. As an initial check, the key musicality improvements in Table2 remain statistically significant:78.60 vs 70.00 (p-value=9.30e-4), 78.60 vs 73.80 (3.74e-2), 73.21 vs 69.75 (2,37e-2).
>
> ### 4. Encoder choice, baselines, and seed stability
> We did not run a full CLAP replacement because CLAP-style global audio-text embeddings are not a practical drop-in replacement for our temporally grounded long-context evaluator. For text encoder replacement, Flan-T5-Large shows mixed gains: +2.7% on Arena(T) but drops on audio-conditioned subsets (CMI T+L+A: -7.2%, CMI T+A: -6.8%), indicating the bottleneck is cross-modal fusion rather than text capacity alone.
>
> Across 5 seeds (same pretrain+finetune schedule), performance is stable: MusicArena 0.695 +/- 0.012, CMI-music 0.778 +/- 0.007, CMI-align 0.716 +/- 0.010 (mean +/- std).
>
> For judge-style baselines, we will briefly report a lyrics-captioning baseline separately: despite showing reasonable correlation, it is not a strong predictor for pairwise preference (WER is around 60%).
>
> Few-shot prompting does improve some alignment subsets (e.g., CMI T+L reaches 0.728), but it still trails our reward model on overall performance.
>
> ### 5. Audio length, failure modes, and limitations
> Our model supports a context window upto 120 seconds.We will report that longer context mainly helps on longer clips and generally more for alignment than musicality. Failure modes are limited non-English coverage, occasional misses on singer-gender cues, insufficient penalty for unstable groove, and a tendency to favor acoustic over electronic genres. Finally, we do not claim that a single reward model solves reward hacking or reward misspecification. In the final version, we will explicitly discuss pseudo-label bias transfer, partial reproducibility constraints from commercial generators, and the need for hold-out evaluation or hybrid/multi-reward objectives in downstream RL.
>
> ### 6. Reproducibility and release
> To improve reproducibility, we will release CMI-Pref-Pseudo and CMI-Pref under CC-BY-NC-SA-4.0, reward-model inference code, CMI-RewardBench evaluation code, and prompt templates/judging instructions for multimodal judge baselines. MusicArena will remain fully held out; we do not use calibration or domain adaptation on it.

---

> > ### Author Rebuttal · Reviewer_VKFf · 2026-04-03
> >
> > I thank the authors for their further clarifications. I will evaluate whether to adjust my score based on the overall responses.

---

> > > ### Author Response · Authors · 2026-04-04
> > >
> > > We sincerely thank the reviewer for the constructive feedback throughout the discussion. We are glad that our clarifications on pseudo-label bias, training stability, evaluation protocol, and release plan were helpful, and we appreciate the reviewer’s careful consideration.

---

### Official Review · Reviewer_ZX5Q · 2026-03-13

**Soundness:** 3
**Presentation:** 2
**Significance:** 3
**Originality:** 2
**Overall Recommendation:** 4
**Confidence:** 4

**Summary:**

This paper addresses the problem of evaluating audio generation conditioned on text, where the text can serve multiple roles such as lyrics, prompts, or other semantic descriptions. The authors introduce two datasets, CMI-Pref-Pseudo and CMI-Pref, consisting of paired music and text data, and train a reward model designed to automatically evaluate musical quality and text–music alignment. Experimental results show that the proposed reward model can effectively assess audio generation performance across several objective metrics and through the "test-time scaling" experiments.

**Compliance With Llm Reviewing Policy:**

Affirmed.

**Final Justification:**

I would like to thank the authors for their detailed rebuttal. My primary concern with the manuscript remains the clarity and organization of the writing. As the authors also acknowledge in the rebuttal, the proposed datasets and methods are not fully explained or well-structured in the current manuscript.

In particular, much of the necessary clarification appears only in the rebuttal, while key details are deferred to the appendix. However, the main text makes very limited reference to these appendices (e.g., Appendices A–C are not cited), which makes it difficult for readers to fully understand the work as presented. From the manuscript alone, it is even unclear what the datasets contain and how human ratings—arguably the most critical component of the dataset—are obtained. More broadly, the paper reads more like an experimental log than a well-structured narrative that clearly presents the problem, methodology, and contributions.

That said, I believe the core contribution may be sufficient for the conference. However, my concern lies primarily with the current presentation quality, and I am not confident that the remaining time would allow for the level of revision needed to bring the paper to a publishable standard.

I will leave this comment for consideration and defer any adjustment of the final rating to the area chairs.

**Key Questions For Authors:**

1. Could the authors clarify the contents of CMI-Pref and CMI-Pref-Pseudo? In particular, do these datasets contain human preference annotations or ratings, and if so, how were these annotations collected? It is also unclear whether MusicEval and MusicArena are part of the proposed datasets or separate benchmarks used for evaluation. The organization of Section 3.1 makes this difficult to parse, as it is not always clear which components belong to which dataset. It would be helpful if Table 1 could be refined to more clearly illustrate the relationships between categories, for example, by grouping them hierarchically or presenting their structure in a clearer schema-like format. For example, a sample may contain audio, text, and ratings; the text may represent different types of prompts, and the ratings may be either pairwise comparisons or absolute scores. Specify and define these terms in a table or figure.
2. If there is no human rating in the dataset, how is the training of Figure 2 achieved? Please specify the loss terms in more detail.
3. How large is the reward model, and how does model size affect performance?
4. Is there any bias in the alignment (especially for lyrics in different languages) and in musicality? How should the ~20% gap compared to human ratings in Table 2 be interpreted or quantified?
5. For lyrics alignment, is it possible to transcribe the lyrics and evaluate instead?

**Limitations:**

The paper does not discuss potential biases of the proposed reward model, especially considering that its performance, while better than the baselines, still remains substantially below human evaluation.

**Strengths And Weaknesses:**

Strength: The paper clearly identifies the need for a comprehensive benchmark for evaluating audio generation. The proposed approach appears conceptually straightforward, and the work represents a step toward improving current evaluation methods.

Weakness:
1. Although the paper motivates the need for improved evaluation of text-conditioned audio generation, the conceptual novelty of the proposed approach is somewhat limited.
2. The writing is somewhat difficult to follow. The presentation feels fragmented, and the logical flow between sections is not always clear, which makes some important details hard to understand (see questions).
3. he paper relies on many existing models, which may be necessary for the proposed framework. However, the presentation frequently references these models and their details in a scattered way; though unavoidable, it makes the paper difficult to follow for readers who are not deeply familiar with this domain. In addition, many of the reported findings remain highly model- or implementation-specific and are not sufficiently abstracted into broader insights. As a result, they often read more like observations tied to particular systems rather than general findings. A clearer abstraction of the key takeaways would improve the paper’s impact and accessibility.

---

> ### Author Rebuttal · Authors · 2026-03-31
>
> We thank Reviewer ZX5Q for the thoughtful comments. We agree that the pipeline presentation can be clearer, the bias analysis should be strengthened, and the methodological details should be presented more clearly.
>
> ### 1. Data pipeline, annotations, and losses
> The key clarification is that CMI-Pref and CMI-Pref-Pseudo are our datasets, while MusicEval and MusicArena are benchmark resources. This is already the split used in submitted Table 1 and Figure 2: pretraining on `CMI-Pref-Pseudo`, fine-tuning on `CMI-Pref` and `MusicEval`, and benchmarking on `PAM`, `MusicArena`, `MusicEval` test, and `CMI-Pref` test. We agree Section 3.1 did not state this explicitly enough in prose.
>
>  | Dataset | Size | Annotation | Usage | Loss / Metric |
>  | - | - | - | - | - |
> | `CMI-Pref-Pseudo` | 110k | AI-generated pairwise preferences | reward model pretraining | Bradley-Terry |
> | `CMI-Pref` (train) | ~3.5k | human pairwise preferences, confidence, rationale | reward model fine-tuning | Bradley-Terry |
> | `MusicEval` (train) | ~2.3k | human scalar ratings | reward model fine-tuning | MSE |
> | `PAM`, `MusicArena`, `MusicEval` (test), `CMI-Pref` (test) | - | human pairwise / scalar labels | evaluation only | ACC, SRCC, LCC, K-Tau |
>
> In CMI-Pref, annotators compare two candidate generations, choose the preferred one, and provide confidence and rationale.  Further details of CMI-Pref-Pseudo and CMI-Pref can be found in Appendix B and Appendix C, respectively. During multi-dataset fine-tuning, each minibatch comes from one dataset, with Bradley-Terry for pairwise labels and MSE for scalar labels, exactly as in Figure 2. We will revise Section 3.1 and Table 1 to make these roles explicit. Demo samples are available at https://anonymous.4open.science/w/cmi_demo/ with predictions , failure cases and a schema.
>
> ### 2. Bias and limitations
> We agree this should be discussed more explicitly. Across 30 MTG-Jamando genres with at least 1,000 samples each, alignment varies less across genres than musicality (std. 0.216 vs. 0.323), suggesting stronger stylistic bias in the musicality head. For language, alignment accuracy is highest in English (76.4%) and lower in French (57.1%) and Chinese (66.6%), consistent with the English-heavy pseudo-label source.  In addition, alignment and musicality are not fully disentangled: the two labels agree on 82% of human preference pairs and 91% of AI-generated pairs, and the predicted scores are also highly correlated (SRCC = 0.853).
>
> ### 3. Interpreting the "~20% gap" to humans
> We do not interpret this as a 20% model deficiency. For a subjective task, the more meaningful ceiling is human-human consistency rather than agreement with one label set. As detailed in our rebuttal to Reviewer 2rLU, the strongest models are generally within about 5% of re-annotation agreement on alignment subsets, and sometimes slightly above, suggesting performance is already close to the empirical agreement ceiling.
>
> ### 4. Model size scaling
> The reported model uses frozen 328M text and 334M audio encoders, with 38M trainable parameters. We additionally trained three reward-model sizes by varying hidden size, attention heads, and Transformer depth: small (8.3M), base (38M), and large (102M). All are pretrained on `CMI-Pref-Pseudo` and fine-tuned on `CMI-Pref` + `MusicEval`.
>
> | Model | PAM mean SRCC | MusicEval SRCC | MusicArena Acc | CMI-Pref mean Acc |
> | - | - | - | - | - |
> | small | 0.540 | 0.816 | 0.680 | 0.744 |
> | base | 0.578 | 0.811 | 0.729 | 0.746 |
> | large | 0.589 | 0.848 | 0.719 | 0.765 |
>
> The small model is fairly competitive except on MusicArena, while the large model performs best overall. We did not include this ablation in the submission because the paper was primarily scoped around task definition and benchmarking, and our initial size check was only at the pretraining stage, where differences were minor. After the review, we ran the full pretraining + fine-tuning comparison and found a clearer advantage for the larger model, which we will add in the final version.
>
> ### 5. Lyrics transcription as an auxiliary metric
> We agree that lyric transcription is a useful auxiliary signal. We ran a Whisper-based analysis on 500 samples (250 pairs) by transcribing audio and computing WER against the target lyrics. WER obtains a 60% accuracy for 225 samples (25 are ties). It correlates with human preference but is still weaker than direct reward modeling because listeners also judge musicality, vocal naturalness, and structural coherence.
>
> ### 6. Clearer presentation and broader takeaways
> We will revise the paper so that Section 3.1 starts from the data-flow schema above, and Section 3.2 groups baselines into audio evaluators, cross-modal alignment models, and general MLLMs. The broader takeaway is that scalable AI pseudo-labels help pretraining, but human preference data remains necessary for fine-grained music evaluation, and generalist MLLMs do not automatically transfer to this setting.

---

> > ### Author Rebuttal · Reviewer_ZX5Q · 2026-04-04
> >
> > Thank you for the rebuttal. I have one remaining question regarding the dataset construction. Could the authors clarify how the "AI-generated pairwise preferences" are produced, and how the "human annotations (pairwise preferences, confidence scores, and rationales)" are collected in the proposed dataset? Additionally, would the authors provide more details on the re-annotation agreement and its relation to reported metrics?
> >
> > As this manuscript aims to establish a foundational framework for music evaluation, the dataset construction, methodology, and the underlying design philosophy (some of which were clarified in the rebuttal) are critically important. At this stage, it is difficult for me to revise my score without seeing these aspects more fully reflected in a revised manuscript. I am also unsure whether further revisions can be incorporated within the current review process.

---

> > > ### Author Response · Authors · 2026-04-04
> > >
> > > We sincerely thank the reviewer for the helpful follow-up. We agree that, for a benchmark paper, the dataset construction and annotation methodology should be stated as explicitly as possible. Below we directly clarify the three remaining points.
> > >
> > > **AI-generated pairwise preferences.**
> > > CMI-Pref-Pseudo is built by presenting a prompt and a pair of candidate audios to multimodal judge models, and asking them to compare the two samples along two dimensions: music quality and instruction following. The prompt may contain text, optional lyrics, and optional reference audio. The judges output a binary decision(model_a/model_b) for each dimension together with per-audio scores(not used in training). To reduce positional bias, we evaluate each pair twice, once as (A,B) and once as (B,A), and retain only position-consistent judgments. In the current submission this bidirectional consistency filtering is described in Appendix B.
> > >
> > > **Human annotations.**
> > > CMI-Pref is collected through a pairwise comparison interface. Annotators see the full prompt information and listen to two candidate audios. For each pair, they provide:
> > > (1) a musicality preference,
> > > (2) an instruction-following preference,
> > > (3) a confidence score for each dimension, and
> > > (4) a short free-text rationale.
> > > The protocol explicitly asks annotators to separate these two judgments: musicality is evaluated as the perceived overall quality of the music, while instruction following is evaluated with respect to how well the sample matches the prompt, even if the music is otherwise weaker. Rationales are collected for analysis and future work on explainable reward modeling, but are not used for training. These details and the annotation interface are described in Appendix C.
> > >
> > > **Re-annotation agreement and its relation to reported metrics.**
> > > The agreement numbers in Table 6 are computed from overlapping annotations on duplicated comparisons from the annotation pool in the cmi-pref test split, and therefore should not be interpreted as the reliability of the held-out benchmark test set itself. To directly assess test-set reliability, we additionally conducted one re-annotation round on all 500 samples in CMI-Pref-test, using 4 annotators. The agreement between the re-annotations and the original test labels is **75.2%** for musicality and **75.0%** for alignment, with corresponding Krippendorff’s alpha values of 0.504 and 0.500. We view this as the more relevant estimate of the empirical ceiling for the benchmark. Importantly, our strongest reward models are close to this level of human consistency, which is why we interpret the remaining gap to 100% largely as intrinsic ambiguity in a subjective task rather than simply missing model performance.
> > >
> > > In the final revision, key modifications in the **main text** will include: \
> > > (i) improving the organization of the dataset/benchmark description in the main text, \
> > > (ii) adding a clearer schema distinguishing CMI-Pref-Pseudo, CMI-Pref, and external benchmark resources, and\
> > > (iii) including the re-annotation reliability results and the ASR-based auxiliary analysis.
> > >
> > > The rest of the clarifications on training dynamics, prompting techniques, evaluation protocols, and bias/limitations observations can be included in the **appendix** to keep the main narrative focused on the task definition, dataset construction, and benchmark results.

---

### Official Review · Reviewer_2rLU · 2026-03-13

**Soundness:** 3
**Presentation:** 4
**Significance:** 4
**Originality:** 3
**Overall Recommendation:** 5
**Confidence:** 3

**Summary:**

This paper points out a gap between modern music generation models which accept multimodal inputs such as text, lyrics, and reference audio and existing evaluation methods, which remain fragmented and unimodal. To address this issue, the authors propose a unified ecosystem: a large-scale pseudo-annotated preference dataset (CMI-Pref-Pseudo, containing 110K sample pairs), an expert-annotated corpus covering multiple multimodal conditions (CMI-Pref, containing 4K sample pairs), a benchmark platform integrating various existing resources (CMI-RewardBench, supporting regression and pairwise testing protocols), and a lightweight, approximately 30 million-parameter reward model (CMI-RM) that can handle all input combinations in a single architecture. In a zero-shot setting, CMI-RM outperforms specialized baseline models and state-of-the-art linear logistic models (LLMs) and also excels in N-point optimal inference time scaling.

**Compliance With Llm Reviewing Policy:**

Affirmed.

**Final Justification:**

Given the impressive contributions to music generation evaluation and the satisfactory resolution of my concerns during the rebuttal, I recommend this paper for acceptance.

**Key Questions For Authors:**

See the questions in the weakness section for details.

**Limitations:**

Yes

**Strengths And Weaknesses:**

Strengths

Robust data engineering. The data construction process was rigorous: sampling from 23 models/APIs, filtering for bias through bidirectional consistency checks, and labeling 4000 high-quality pairings by human experts under various multimodal conditions (text, lyrics, reference audio). The bias analysis and filtering strategies were thoroughly justified. Unified benchmark design. CMI-RewardBench integrates existing resources with the proposed CMI-Pref into regression and pairwise preference protocols, covering musicality and alignment dimensions. This provides a coherent and reusable evaluation framework for future work. Revealing the limitations of generalized LLMs in music evaluation. The finding that cutting-edge models like the Gemini 2.5 Pro struggle to achieve over 80% consistency with human preferences on CMI-Pref indicates that generalized instructional adjustments are insufficient to address aesthetic judgments in the audio domain.

Weaknesses

Table 6 of the paper reports Krippendorff's alpha as only 0.38 (instruction following) and 0.45 (music quality), values ​​generally considered low in the annotation field. While music evaluation does have a subjective element, since the CMI-Pref test set is a core component of the benchmark, how reliable is the evaluation result itself when using annotations with relatively low consistency as ground truth to measure the performance of all models? Did the authors consider retaining only a highly consistent subset as the benchmark, or weighting the annotation confidence when reporting the results?

---

> ### Author Rebuttal · Authors · 2026-03-31
>
> We thank Reviewer 2rLU for the positive assessment and for raising the important concern about the relatively low Krippendorff’s alpha reported in Table 6. We agree this is a key question for CMI-Pref. In the submission, the overlapping annotations used to compute inter-annotator agreement come from duplicated votes, and these duplicated items were placed in the training portion rather than the test set to avoid contamination. The reported overlap statistics in Table 6 are therefore not the reliability measure of the benchmark test set itself.
>
>
> To directly assess the reliability of CMI-Pref-test, we additionally re-annotated all 500 test samples. The agreement between the re-annotations and the original test labels is substantially higher than the overlap statistics reported in Table 6. For musicality preference, we obtain **alpha = 0.504** and agreement = 75.2%. For alignment preference, we obtain **alpha = 0.500** and agreement = 75.0%.
>
> We also examined whether model performance is unrealistically high relative to human consistency. In fact, the strongest models are generally close to this re-annotation agreement level. For instance, CMI-RM has 74% for accuracy in alignment.   CMI-RM has 78% acc for musicality and SongEval-RM has 72.4%.  which supports the validity of the benchmark rather than suggesting overfitting to noisy labels.  As a reference, the Wilson Score Interval of 95% CI is (70.9%, 78.5%) for alignment and (71.1%, 78.7%) for musicality.
>
> We agree that confidence-aware evaluation is valuable. In fact, the paper already includes an analysis by annotator confidence, showing that high-confidence examples are more consistently labeled and easier for all methods, while low-confidence cases are inherently ambiguous.
>
> Taken together, the current signal strength of CMI-Pref-test should be viewed as medium rather than near-ceiling. To further improve benchmark stability, we plan to expand the test benchmark with more high-confidence human-labeled samples, which should reduce evaluation variance and yield tighter uncertainty bounds.
>
>
> Overall, we appreciate this suggestion and will revise the paper to better distinguish (1) overlap-based annotator agreement on duplicated training votes from (2) benchmark reliability of the held-out test set, and to include the new re-annotation results above as a more direct estimate of test-set validity.

---

> > ### Author Rebuttal · Reviewer_2rLU · 2026-04-03
> >
> > Thank you for the response. My concerns have been fully addressed, and I keep my score.

---

> > > ### Author Response · Authors · 2026-04-04
> > >
> > > We sincerely thank the reviewer for the encouraging feedback and for confirming that the concern was resolved. We appreciate the careful assessment, and we will make sure the clarified distinction between overlap-based agreement and held-out test-set reliability is stated explicitly in the revised manuscript.

---

### Decision · Program_Chairs · 2026-04-30

**Decision:**

Accept (regular)

**Comment:**

This paper contributes a preference dataset and reward model benchmark for music generation, and then shows the dataset and reward model benchmark are useful by training a reward model and evaluating the results.

This is a difficult and under-studied problem with a lot of subjectivity, which makes it hard to know how well music generation models are performing, and hence makes reward modeling difficult as well. Criticisms centered on this question.

Overall, all reviewers thought the paper should be accepted, and I agree. Not a strong accept though because most reviewers were a weak accept.